# Differentiable Decision Tree via "ReLU+Argmin" Reformulation

**Qiangqiang Mao, Jiayang Ren, Yixiu Wang, Chenxuanyin Zou, Jingjing Zheng, Yankai Cao**[*]
University of British Columbia, Vancouver, Canada

## Abstract

Decision tree, despite its unmatched interpretability and lightweight structure, faces two key issues that limit its broader applicability: non-differentiability and low testing accuracy. This study addresses these issues by developing a differentiable oblique tree that optimizes the entire tree using gradient-based optimization. We propose an exact reformulation of hard-split trees based on "ReLU+Argmin" mechanism, and then cast the reformulated tree training as an unconstrained optimization task. The ReLU-based sample branching, expressed as exact-zero or non-zero values, preserve a unique decision path, in contrast to soft decision trees with probabilistic routing. The subsequent Argmin operation identifies the unique zero-violation path, enabling deterministic predictions. For effective gradient flow, we approximate Argmin behaviors by scaling softmin function. To ameliorate numerical instability, we propose a warm-start annealing scheme that solves multiple optimization tasks with increasingly accurate approximations. This reformulation alongside distributed GPU parallelism offers strong scalability, supporting 12-depth tree even on million-scale datasets where most baselines fail. Extensive experiments demonstrate that our optimized tree achieves a superior testing accuracy against 14 baselines, including an average improvement of 7.54% over `CART`.

## 1  Introduction

Decision trees have attracted significant attention in machine learning primarily due to their interpretability and lightweight structures. Their strength lies in transparent "IF-THEN" rules, making them ideal for tasks requiring clear decision-making processes. However, their practical use is often limited by the lower test accuracy, forcing a shift towards other models that sacrifice interpretability advantages. Another limitation is the non-differentiability, which severely restricts their gradient-based applications, especially when embedded in optimization tasks. For instance, in reinforcement learning, decision trees offer an interpretable alternative to neural networks for policy representation; yet, non-differentiability remains a major barrier [Vos and Verwer, 2024, Marton et al., 2025].

Aiming for higher accuracy and fewer parameters, oblique decision tree, a pivot extension of classic orthogonal tree, holds great potential. Oblique tree uses linear feature combination for hyperplane splits. When data distribution follows hyperplane boundaries, it tends to generate smaller trees with higher accuracy [Costa and Pedreira, 2023]. Nevertheless, inducing oblique trees is challenging, owing to innumerable linear combinations at each node [Zhu et al., 2020]. Earlier works focus on optimizing splits at an individual node using greedy algorithms like `CART-LC` [Breiman et al., 1984] and `OC1` [Murthy et al., 1994]. Alternative methods rely on greedy orthogonal `CART` to induce oblique trees by rotating feature space, exemplified by `HHCART` [Wickramarachchi et al., 2015] and `RandCART` [Blaser and Fryzlewicz, 2016]. Despite their advancements, such greedy methods may be suboptimal due to weaker splits at subsequent nodes. To avoid this, non-greedy approach `TAO` [Zharmagambetov and Carreira-Perpinan, 2020] optimizes a subset of nodes at each step. Considering optimizing all

---

[*]Corresponding author: yankai.cao@ubc.ca.

nodes, Bertsimas and Dunn [2017] presents optimal tree to formulate tree training as a mixed-integer programming (MIP). Besides, sparse optimal trees [Lin et al., 2022] complements MIP-based work by considering sparsity. However, these methods often face scalability issues, especially for oblique tree with expanded search space. Recent efforts in optimal oblique tree [Boutilier et al., 2023, Zhu et al., 2020] have been confined to classification tasks with limited categorical predictions. In contrast, regression tasks with infinite continuous outputs remain challenging. In response, the originators of MIP-based work propose a local search alternative `ORT-LS` [Dunn, 2018] for tasks that are unsolvable by MIP. However, `ORT-LS` still suffers from high computational cost and suboptimal accuracy.

To significantly improve computational tractability compared to those MIP formulations, in this work, we reformulate entire tree training as an unconstrained optimization task. Our reformulation makes it easily solvable via gradient-based tools and also facilitates adapting it into a differentiable tree. The classic tree's non-differentiability mainly arises from two sources of hard decisions: hard splits at branch nodes that determine sample branching, and the unique decision path that assign each sample to a specific leaf node for sample prediction. Given the non-differentiability of hard decisions, two intuitive solutions have been used: treating the gradient via straight-through estimators (STE) like `DGT` [Karthikeyan et al., 2022], `GradTree` [Marton et al., 2023] and `DTSemNet` regression tree [Panda et al., 2024], and approximating binary decisions (0 or 1) with continuous probabilities in (0,1) using soft approximations like sigmoid function in soft trees [İrsoy et al., 2012, Wan et al., 2021, Frosst and Hinton, 2017] and other soft variants like smooth-step functions in `TEL` [Hazimeh et al., 2020]. However, STE may neglect critical gradient information, resulting in suboptimal learning, as evidenced by results reported in the original work of [Marton et al., 2023], where their method underperforms `CART` on 17 of 36 datasets (their Tables 1 and 2), as well as in our own experiments. Regarding soft approximations, previous efforts predominantly construct "soft" decision trees with soft splits, probabilistic path and predictions. Nonetheless, there do exist scenarios where a hard split or decision is not only appropriate but also imperative, as in Appendix A. Besides, other gradient-based trees exist, but they might not preserve classic tree structures or be exact reformulations, such as `DNDT` [Yang et al., 2018] which uses Kronecker products for class predictions, and the method by Norouzi et al. [2015] which uses a surrogate loss. Due to their differences in structure, formulation and optimization, we consider them complementary work suited for distinct application scenarios.

To retain the two sources of hard decisions, we propose a "ReLU+Argmin"-based exact reformulation. ReLU functions enforce hard decisions to direct samples left or right, and quantify violations of the correct directions that a sample must follow to reach its unique leaf. Our ReLU-based splits, in sharp contrast to soft splits with left-right probability, yield hard (exact-zero or non-zero) decisions for sample branching. This leverages ReLU's property of producing distinct zero and non-zero values at True-False decision boundaries. Such a property has ever been used to implicitly mimic a decision tree using neural networks [Lee and Jaakkola, 2020], in contrast to our direct application for reformulating and optimizing a tree. Next, the correct decision path is identified as the unique path with zero cumulative violations. This can be easily formulated via Argmin operation over zero and non-zero (positive) values. To avoid undefined gradient in Argmin, we approximate it with a scaled softmin function. Noticeably, two clarifications are necessary: first, our Argmin is a mathematical operation over a discrete set, and it should not be confused with "Argmin Differentiation" or "Implicit Differentiation" [Gould et al., 2016]. The latter is used in `LatenTree` [Zantedeschi et al., 2021], which frames tree training as a relaxed MIP problem within a bilevel optimization setting. However, their work is not directly related to our unconstrained task, and unrelated to approximate the Argmin operation itself. Second, our softmin approximation is introduced purely to facilitate gradient backpropagation during training; at the inference phase, our method reverts to using Argmin for deterministic behavior. This technique of using approximation during training while reverting to original function during inference was previously used in the work of [Mao and Cao, 2024].

**Our Contributions:** First, we propose a "ReLU+Argmin"-based exact reformulation for hard-split trees, avoiding softness introduced in sample branching and predictions. Second, we cast the entire tree training as an unconstrained optimization task, introducing a scaled softmin function to approximate Argmin behaviors for effective gradient flow. Third, to balance approximation degree with numerical instabilities, we present a strategy of multi-run warm start annealing to progressively refine solutions. Fourth, our implementation supports multi-GPU acceleration, significantly enhancing scalability. Finally, we provide an extensible "**R**eLU+**A**rgmin"-based **D**ifferentiable **D**ecision **T**ree optimization framework (termed `RADDT`) for inducing both regression trees and classification trees. The source code is available in `https://github.com/YankaiGroup/RADDT`.

**Performance:** Experiments mainly focus on regression tasks across 17 medium-scale datasets and 7 million-scale datasets, showing our competitive testing accuracy as well as strong scalability. To enable a more convincing comparison, additional evaluations on the same datasets used in the original work of certain compared baselines further highlight the superiority of our method.

- Our optimized trees outperform the compared decision trees in testing accuracy. Notably, it outperforms `CART` by 7.54%, local search `ORT-LS` by 3.72%, and gradient-based `LatentTree` by 6.24%, and `DGT` by 2.66% on average.
- Our trees with linear predictions impressively outperforms tree ensembles, including random forest by 2.01%, `XGBoost` by 1.12% and gradient-based `TEL` by 0.76%.
- Our method successfully scales to 12-depth deep tree on million-scale datasets, where existing gradient-based trees like `GradTree`, `SoftDt`, `DGT` and `LatentTree` fail.

## 2 Foundations of Oblique Decision Tree

We explore oblique decision trees from an optimization perspective by formulating tree training as an optimization problem for both regression and classification tasks. For ease of understanding, we follow the notation for optimal binary trees as used in the original work of Bertsimas and Dunn [2017]. Consider a dataset $\{\boldsymbol{x}_i, y_i\}_{i=1}^n$ with input $\boldsymbol{x}_i \in [0,1]^p$, and output values $y_i \in [0,1]$ for regression and $y_i \in \{1, \cdots, c\}$ for classification with $c$ classes. A binary tree of depth $D$ comprises $T = 2^{D+1} - 1$ nodes, where each node is indexed by $t \in \mathbb{T} = \{1, \cdots, T\}$ in a breadth-first order. The nodes can be categorized into two types: branch nodes, which execute branching tests and are denoted by $t \in \mathbb{T}_B = \{1, \cdots, \lfloor T/2 \rfloor\}$, and leaf nodes denoted by $t \in \mathbb{T}_L = \{\lfloor T/2 \rfloor + 1, \cdots, T\}$, responsible for leaf predictions. Each branch node comprises a split weight $\boldsymbol{a}_t \in \mathbb{R}^p$ and a split threshold $b_t \in \mathbb{R}$ to conduct a branching test ($\boldsymbol{a}_t^T \boldsymbol{x}_i \leq b_t$) for samples allocated to that particular branch node. If a sample $\boldsymbol{x}_i$ passes the branching test, it is directed to the left child node at index $2t$; otherwise, to the right at index $2t+1$. Each leaf node contains $\theta_t$ to provide a prediction value specific to the current leaf, which varies depending on the task. For classification, $\theta_t = \{h_t \in \mathbb{R}^c\}$, and the prediction for a sample $\boldsymbol{x}_i$ assigned to leaf $t$ is $\hat{y}_i = h_t$. For regression, $\theta_t = \{\boldsymbol{k}_t \in \mathbb{R}^p, h_t \in \mathbb{R}\}$, which considers two types of leaf prediction: linear and constant prediction. Specifically, tree with linear predictions involves a linear feature combination [Quinlan, 1998], with a general form of $\hat{y}_i = \boldsymbol{k}_t^T \boldsymbol{x}_i + h_t$. Tree with constant predictions is a special case of linear predictions, where $\boldsymbol{K}$ remains zero. It is the most commonly used type in existing decision tree methods, with $\hat{y}_i = h_t$. The training of oblique trees involves solving the following optimization problem:

$$\min_{\boldsymbol{A}, \boldsymbol{b}, \theta} \ \ell\left(y_i, \hat{y}_i\right) \tag{1a}$$

$$\text{s.t. } \hat{y}_i = f_{tree}(\boldsymbol{A}, \boldsymbol{b}, \theta, \boldsymbol{x}_i), \ i \in \{1, \cdots, n\}, \tag{1b}$$

where $f_{tree}$ is the decision tree model, $\ell\left(\cdot\right)$ is square error loss for regression and cross entropy loss for classification. $\boldsymbol{A} = \{\boldsymbol{a}_1, \cdots, \boldsymbol{a}_{\lfloor T/2 \rfloor}\}$ and $\boldsymbol{b} = \{b_1, \cdots, b_{\lfloor T/2 \rfloor}\}$ are tree split parameters, $\theta = \{\theta_{\lfloor T/2 \rfloor + 1}, \cdots, \theta_T\}$ is leaf prediction parameter.

## 3 Unconstrained Optimization Reformulation for Oblique Tree Training

We first propose an exact reformulation of decision trees and further cast the training as an unconstrained task, allowing for gradient-based optimization for improved solvability and accuracy.

### 3.1 "ReLU+Argmin"-based exact reformulation for hard-split decision trees

**ReLU-based hard splits and correctly-directed path formulation** (no softness at branch nodes for sample branching)**:** A sample $\boldsymbol{x}_i$ is correctly assigned to a leaf node $t$ by following a sequence of left-right decisions at its ancestor nodes. The ancestor nodes set $\mathbb{A}_t$ consists of left $\mathbb{A}_t^l$ and right $\mathbb{A}_t^r$ ancestors, traversed via left and right branch, respectively, such that $\mathbb{A}_t = \mathbb{A}_t^l \cup \mathbb{A}_t^r$. For left ancestor node $j \in \mathbb{A}_t^l$, $\boldsymbol{x}_i$ must pass the branching test $\boldsymbol{a}_j^T \boldsymbol{x}_i \leq b_j$ to be directed to the left, whereas for right $j \in \mathbb{A}_t^r$, it must fail the test to proceed right. To formulate these left-right hard decisions for $\boldsymbol{x}_i$, we use ReLU functions to characterize the violation of correct direction at each node $j$, termed $v_{i,j}$:

$$v_{i,j} = \begin{cases} ReLU\left(\boldsymbol{a}_j^T \boldsymbol{x}_i - b_j\right), & j \in \mathbb{A}_t^l \\ ReLU\left(b_j - \boldsymbol{a}_j^T \boldsymbol{x}_i\right), & j \in \mathbb{A}_t^r, \end{cases} \tag{2}$$

where $v_{i,j} = 0$ indicates that $\boldsymbol{x}_i$ follows the correct direction, i.e. left for $\mathbb{A}_t^l$ and right for $\mathbb{A}_t^r$, while $v_{i,j} \neq 0$ is a violation of correct direction. These deterministic ReLU-based zero and non-zero values eliminate the softness in sample branching, as opposed to soft approximations with continuous probabilities in (0,1) at each branch node. The cumulative violations across all branch nodes are:

$$U_{i,t} = \sum_{j \in \mathbb{A}_t^l} v_{i,j} + \sum_{j \in \mathbb{A}_t^r} v_{i,j}, \tag{3}$$

where $U_{i,t} = 0$ indicates $\boldsymbol{x}_i$ follows all correct directions to leaf node $t$; otherwise $U_{i,t} \neq 0 (> 0)$ corresponds to the overall violations of correct sample assignments.

**Unique decision path formulation using Argmin** (no softness at leaf nodes for sample prediction)**:** Since $x_i$ follows a unique decision path to a specific leaf node, only one leaf node holds a violation-free path. Such a unique zero $U_{i,t}$ removes the softness in sample assignment and prediction, in contrast to soft approximations with probabilistic paths for predictions. The unique path can be easily formulated by Argmin function (more concretely, one-hot encoded Argmin $\mathbb{M}$) applied to $U_{i,t}$:

$$\mathbb{M}(U_{i,t}) = \mathbb{1}\left(t = \left(\text{Argmin}(\mathbf{U}_i) + \lceil T/2 \rceil\right)\right), \tag{4}$$

where $\mathbf{U}_i = \{U_{i,\lfloor T/2 \rfloor+1}, \cdots, U_{i,T}\}$. $\mathbb{M}(\cdot)$ is the one-hot encoding of the result of Argmin operation, only outputting one at leaf node $t$ with $U_{i,t} = 0$; otherwise outputting zero.

For a clearer understanding of our reformulations, a 2-depth tree example is provided in Figure 1. To be correctly assigned to leaf node 5, the sample must pass the branching test at nodes 1 with $\boldsymbol{a}_1^T \boldsymbol{x}_i \leq b_1$ to be directed to left, which implies that $ReLU\left(\boldsymbol{a}_1^T \boldsymbol{x}_i - b_1\right) = 0$. The sample also needs to fail at node 2 to be directed to right with $\boldsymbol{a}_2^T \boldsymbol{x}_i > b_2$, which implies that $ReLU\left(b_2 - \boldsymbol{a}_2^T \boldsymbol{x}_i\right) = 0$. The corresponding violations are $v_{i,1} = 0$ and $v_{i,2} = 0$, leading to a unique zero-violation path with $U_{i,5} = v_{i,1} + v_{i,2}$. For other leaf nodes, the total violations are non-zero (positive). The unique $\mathbb{M}(U_{i,5}) = 1$ indicates $\boldsymbol{x}_i$ is correctly assigned to leaf 5.

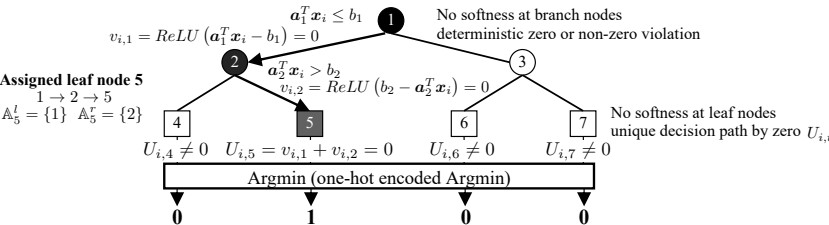

Figure 1: Illustrative example for a unique path $1 \to 2 \to 5$ of "ReLU+Argmin"-based hard tree.

## 3.2 Unconstrained optimization for "ReLU+Argmin"-based tree training

Next, we cast reformulated-tree training as an unconstrained optimization task with objective $\mathcal{L}$ as:

$$\mathcal{L} = \sum_{i=1}^{n} \sum_{t \in \mathbb{T}_L} \mathbb{M}(U_{i,t}) \ell(y_i, \hat{y}_i). \tag{5}$$

**Differentiability analysis of ReLU and Argmin:** The trainable variables $\boldsymbol{A}$, $\boldsymbol{b}$, $\theta$ are expressed in $v_{i,j}$ and $U_{i,t}$, where ReLU and Argmin are involved. The gradient behavior at ReLU(0) has been extensively studied and is considered theoretically negligible, though numerical effects may arise due to float-point precision [Bertoin et al., 2021]. Consistent with this, we treat it as inconsequential for gradient-based optimization, as evidenced by the success of ReLU networks [Leonardi and Spallanzani, 2020]. Regarding Argmin, to avoid undefined gradient flow at certain points, we use a scaled softmin function to approximate Argmin behaviors in Equation (4).

**Scaled softmin to approximate Argmin:** Scaled softmin $\mathbb{S}(\cdot)$ is applied to $U_{i,t}$ by Equation (6). Under extreme scaling, softmin ensures that zero-valued $U_{i,t}$ approaches one; otherwise to zero. The scale factor $\alpha$ plays a critical balance between approximation accuracy and optimization stability. Regarding $\alpha$ selection, a larger $\alpha$ yields a closer approximation, but may cause numerical instability, potentially compromising optimization capabilities. Further empirical analyses are given in Appendix B. Identifying optimal $\alpha$ that balances approximation degree and differentiability remains a challenge. To mitigate this, we propose an annealing strategy in Section 4.1 for a better trade-off.

$$\mathbb{S}(U_{i,t}) = \frac{e^{\alpha(-U_{i,t})}}{\sum_{t \in \mathbb{T}_L} e^{\alpha(-U_{i,t})}}. \tag{6}$$

Notably, this softmin operation is used only during training to facilitate gradient backpropagation. In inference, $\theta$ is deterministically recalculated for $\hat{y}_i$, as described in Appendix C. More importantly, it does not introduce softness into sample branching and prediction, as the zero-valued ReLU outputs and violation-free path described in Section 3.1 are established prior to the approximation step.

Following Equation (6), we redefine the optimization task $\mathcal{L}$ in Equation (7).

$$\mathcal{L} = \sum_{i=1}^{n} \sum_{t \in \mathbb{T}_L} \mathbb{S}\left(U_{i,t}\right) \ell\left(y_i, \hat{y}_i\right). \tag{7}$$

## 4 Differentiable Tree Training via Gradient-based Entire Tree Optimization

Our reformulated task in Equation (7) closely approximates non-differentiable tree training problem, which can be efficiently solved by our proposed entire tree optimization framework.

### 4.1 Multi-run warm start annealing strategy for scaling softmin operations

Regarding the challenge of selecting an optimal $\alpha$ as earlier discussed for Equation (6), we propose a strategy of multi-run warm start annealing to refine solutions through multiple optimization tasks with increasingly accurate approximations. The annealing has been ever used [Hehn and Hamprecht, 2017, Karthikeyan et al., 2022, Lee and Jaakkola, 2020] to balance the trade-off, though not specifically for our application to softmin. However, their annealing is typically applied within a single training run, such as changing $\alpha$ every few epochs, which may affect stable gradient updates. In contrast, our annealing operates across multiple runs. The solution from an optimization task with a smaller $\alpha$ is used to warm-start the next task with a larger $\alpha$. By starting with a smaller $\alpha$ and gradually increasing it, this enhances approximation accuracy, while mitigating numerical instability associated with larger $\alpha$. Detailed steps including sampling of scaled factors $\{\alpha_1, \cdots, \alpha_n\}$ are given in Appendix D, and the implementation is integrated within our tree optimization framework, Appendix G, Algorithm 1. For the $\alpha$ selection, although binary search provides a structured way to explore different $\alpha$ values by evaluating the performance at the midpoint and adjusting $\alpha$ boundaries accordingly, it still involves testing isolated $\alpha$ values without leveraging the solutions from previous $\alpha$ trials. Our subsequent ablation study shows that gradually increasing $\alpha$, rather than a fixed large value, leads to better training outcomes. We detail binary search comparison in Appendix D.

### 4.2 Influence of initial values and their adjustments for gradient-based optimization

A good initialization is crucial. Since softmin operates on $U_{i,t}$, which summarizes ReLU-based violations that are either zero or non-zero, samples near decision boundaries ($\boldsymbol{a}_j^T \boldsymbol{x}_i - b_j = 0$) may yield non-zero but near-zero violations. An illustration is given in Appendix E. While such near-zero violations do not affect Argmin behaviors, which only select the unique path with $U_{i,t} = 0$, they can sensitively affect softmin. Specifically, softmin may fail to ideally produce a value close to one, reducing approximation accuracy. A bad initial solution for $\boldsymbol{A}$ and $\boldsymbol{b}$ can result in a poor optimization outcome. To mitigate this, $\boldsymbol{A}$ and $\boldsymbol{b}$ can be adjusted to maximize the margin between the decision boundary and nearest samples, without altering existing sample assignments as illustrated by green line in Appendix E, Figure 3. This adjustment can be easily achieved either by calculating the median of $\boldsymbol{a}_j^T \boldsymbol{x}_i$ for the closest samples on both sides of the boundary, or using support vector machine to treat sample assignments as a binary classification task with explanations in Appendix E.

Noticeably, this strategy mainly handles cases where $\boldsymbol{a}_j^T \boldsymbol{x}_i - b_j \approx 0$. However, it becomes ineffective in the corner case where $\boldsymbol{a}_j = 0$ and $b_j = 0$ with same violations for all leaves. This limitation can impair optimization. To mitigate this, we typically incorporate this strategy with multiple random initializations. Fortunately, our extensive experiments show that such corner cases appear to be extremely rare in practice, and we have not observed noticeable performance degradation.

### 4.3 Differentiable decision tree optimization framework

Unlike greedy methods that optimize each node sequentially, our approach concurrently optimizes the entire tree, including $\boldsymbol{A}$ and $\boldsymbol{b}$ at all branch nodes and $\theta$ at all leaf nodes. Our "**R**eLU+**A**rgmin"-based **D**ifferentiable **D**ecision **T**ree (termed RADDT), outlined in Appendix G, Algorithm 1, begins at

multiple random initialization (*Line 4 - 5*). This increases the chance of finding better solutions. Each initialization is adjusted to achieve the maximum margin (*Line 4*). For each start, the optimization with our annealing is conducted to produce a tree candidate (*Line 6 - 15*). The final optimal tree is selected by comparing candidates from multiple starts, with each evaluated using deterministic leaf predictions and exact loss calculations without approximation (*Line 11 - 13*). This framework is readily implementable using existing tools like PyTorch. Despite the introduction of additional hyperparameters in gradient-based optimization, tuning them is not typically necessary because their effects are straightforward. More empirical analyses are detailed in Appendix F.

# 5   Numerical Experiments and Discussions

We mainly evaluate regression tasks, but also include the same classification datasets used by certain baselines for a more convincing comparison. Our tree with constant or linear predictions, termed the respective `RADDT` and `RADDT-Linear`, is compared against a broad range of trees in testing accuracy, training optimality and training time. The **14 baselines** include: **(a)** greedy `CART`, `HHCART`, `RandCART`, and `OC1`; **(b)** non-greedy `TAO` [Zharmagambetov and Carreira-Perpinan, 2020]; **(c)** gradient-based trees using STE like `GradTree` [Marton et al., 2023], `DGT` [Karthikeyan et al., 2022] and `DTSemNet` [Panda et al., 2024]; **(d)** sigmoid-based soft tree `SoftDT` [Frosst and Hinton, 2017]; **(e)** relaxed MIP-based tree solved via implicit differentiation `LatentTree` [Zantedeschi et al., 2021]; **(f)** local search `ORT-LS` [Dunn, 2018]; **(g)** soft tree variant using smooth-step function `TEL` [Hazimeh et al., 2020] (Tree ensemble); **(h)** other ensembles like random forest `RF` and `XGBoost`. Our primary focus is on comparing oblique trees. The inclusion of orthogonal `CART` serves as a foundational work, aligning with prior work on oblique trees, such as `TAO`, `DGT`, `DTSemNet` and `TEL`, which also use `CART` for comparison. Although outperforming `CART` is an expected outcome, its inclusion helps quantify the amount of improvements achievable by oblique tree. `GradTree` is included to enable a fair comparison against other STE-based oblique trees like `DGT` and `DTSemNet`. Since it remains uncertain whether oblique trees can match tree ensemble's accuracy, we include `RF` and `XGBoost` as a commonly used and strong ensemble baselines, and `TEL` as the oblique ensemble counterpart.

These 14 baseline comparisons are conducted across **four groups of datasets**: **(i)** 17 medium-sized regression datasets with fewer than 41k samples (primary focus); **(ii)** 27 classification and 9 regression datasets used in the papers of certain baselines (for a more credible comparison); **(iii)** 4 real-world datasets with about 100 samples, and 3 synthetic datasets with 5000 samples (for comparing global optimality with global tree `ORT-MIP` [Bertsimas and Dunn, 2017] and known ground truth); **(iv)** 7 large-scale datasets each with at least 1 million samples (for evaluating scalability).

Detailed dataset information and usage are given in Appendix H. Comparisons focus on accuracy in Coefficient of Determination (usually denoted as $R^2$), and computational time in seconds. The Friedman Rank [Sheskin, 2020] is also used to statistically sort the compared methods according to their testing accuracy, with a lower value indicating better performance. Comprehensive details on the implementation for each compared method, and computing facilities are given in Appendix I.

## 5.1   Testing accuracy comparison

To evaluate the testing accuracy of our optimized tree, we compare it against diverse decision trees and 3 tree ensembles with different tree induction methods. For a fair comparison, we conduct depth tuning for all tree methods to select the optimal tree depth from 1 to 12 (except for `CART` from 1 to 100 due to its tendency to rely on deeper depth for higher accuracy).

**Comparison against other decision trees:**  In Table 1, our `RADDT` consistently outperforms compared trees in test accuracy across Group (i) 17 datasets. Specifically, `RADDT` surpasses the foundational `CART` by 7.54%, greedy `HHCART` by 5.64%, and `ORT-LS` by 3.72%. These results underscore the effectiveness of our tree optimization. Besides, Friedman rank comparisons reinforce these findings, with `RADDT` attaining the highest rank. Detailed results for each dataset are provided in Appendix K.1.

Regarding existing gradient-based trees, since `DGT` and `LatentTree` fail to scale to depth 12, we evaluate them under another fixed-depth setting in Section 5.2. `GradTree`, as observed in Table 1, performs poorly in our experiments. This result is within our expectations, as their original paper reports that `GradTree` underperforms `CART` on 17 out of 36 datasets (see their Table 1 and 2 in [Marton et al., 2023]). Nonetheless, to ensure a more credible comparison, we further evaluate our

Table 1: Test accuracy for compared decision trees on 17 medium-sized datasets in Group (i).

| | Greedy Methods | | | | Gradient-based[1, 2] | | Local Search | Ours |
|---|---|---|---|---|---|---|---|---|
| | CART | RandCART | HHCART | OC1 | GradTree | SoftDT | ORT-LS | RADDT |
| Testing Accuracy ($R^2$, %) | 74.85 | 71.20 | 76.75 | 73.31 | 64.30 | 72.72 | 78.67 | **82.39** |
| Tree Depth | 10.24 | 8.12 | 8.29 | 8.12 | 10.29 | 10.29 | 6.24 | 7.29 |
| Friedman Rank | 4.65 | 5.88 | 3.65 | 5.06 | 7.00 | 4.76 | 3.24 | **1.76** |

[1] Other two gradient-based DGT and LatentTree are compared in Table 3 with fixed depths due to scalability issues at depth 12.
[2] More credible results on their originally-used dataset are discussed below, include GradTree, SoftDT, DGT, and DTSemNet.

method on the same classification datasets used in their study. Similar same-dataset comparisons are also conducted for SoftDT, DGT-Linear, DTSemNet and non-greedy TAO-Linear as below.

**A more credible same-dataset comparison on Group (ii) datasets originally used by five baselines:**
For a more convincing comparison, we evaluate on the same datasets used in the original works of five baselines: 27 classification datasets in GradTree, 4 regression datasets in soft tree [İrsoy et al., 2012], and 5 shared regression datasets among DGT-Linear, TAO-Linear and DTSemNet. Dataset information is given in Appendix H. Besides using same datasets, we also compare directly with the results reported in their own papers. For fairness, our method is evaluated using each baseline's experimental setup, including same data preprocessing and evaluation metrics. Full implementation details are given in Appendix K.2. For GradTree, with detailed results for each dataset given in Appendix K.3, Table 15, our method outperforms GradTree's reported testing results by an average of 4.99%. Regarding soft tree, results in Appendix K.4, Table 16, show that RADDT achieves 6.88% higher test accuracy, again aligning with prior trends. Regarding DGT-Linear, TAO-Linear and DTSemNet, the DTSemNet work directly reports TAO-Linear's original results due to the absence of open-source code for TAO-Linear. Since DTSemNet itself is a tree with linear predictions, for consistency, it adapts the original DGT tree with constant predictions to a linear variant DGT-Linear. Following this comparison from DTSemNet, we extend DTSemNet's Table 4 by including our results under same experimental setup. As in Appendix K.5, Table 17, ours outperforms these baselines on 4 out of 5 datasets (except "YearPred"), achieving the best average rank of 1.6, followed by DTSemNet with 2.4, TAO-Linear with 2.6, and DGT-Linear with 3.2, further supporting our previous findings.

**Statistical analysis regarding Table 1:** While Table 1 empirically shows that RADDT is superior to the other compared trees, we further conduct a paired T-test to statistically validate the significant difference among these tree methods. The null hypothesis asserts that RADDT is not significantly different from other trees. If calculated $p$-value is less than tolerance $\tau$, the null hypothesis can be rejected, indicating statistical significance. The t-statistic (black point), $p$-value and 95% confidence are shown in Figure 2. The results for all comparisons with $p < \tau = 0.1$ and positive t-statistics consistently indicate that RADDT is superior to these compared decision trees with statistical significance.

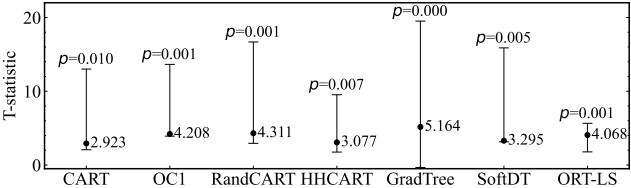

Figure 2: Paired T-test comparing RADDT with other trees (setting significance level $\tau = 0.1$).

**Comparison against tree ensembles on Group (i) 17 datasets:** Following decision tree comparison, we proceed to compare our method with RF, XGBoost and TEL to validate its competitive accuracy. Our RADDT slightly underperforms tree ensembles on average. In contrast, RADDT-Linear outperforms RF by 2.01%, XGBoost by 1.12% and TEL by 0.76%, as shown in Table 2. The superiority of RADDT-Linear is further supported by Friedman rank, where RADDT-Linear ranks highest, matching that of XGBoost. Detailed results for each dataset are provided in Appendix K.1. These findings highlight that our method even performs competitively compared to tree ensembles. For a fair comparison with tree ensembles, this experiment has performed a hyperparameter tuning with 300-combination search, including depth up to 50 and number of trees up to 500, following guidelines from [Oshiro et al., 2012, Probst et al., 2019]. Detailed tuning procedures are given in Appendix J. In addition to this 300-combination parameter tuning for ensembles, we also conduct a more extensive tuning like 10,000-combination search in Appendix K.6. Despite extensive tuning, our key findings remain unchanged, and the statistical Friedman rank shows no significant difference. Importantly, our aim is not to claim superiority over ensembles. Since any base learner can be improved via

ensembling like bagging, comparing a single tree to ensembles is inherently unfair. Instead, we provide a reference showing that our tree can match ensemble's accuracy with far fewer parameters.

Table 2: Test accuracy for our trees and 3 tree ensembles on 17 medium-sized datasets in Group (i).

|  | RF | XGBoost | TEL | RADDT | RADDT-Linear |
|---|---|---|---|---|---|
| Number of Trees | 314.71 | 352.94 | 10 | 1 | 1 |
| Test Accuracy ($R^2$, %) | 82.62 | 83.51 | 83.87 | 82.39 | **84.63** |
| Friedman Rank | 2.88 | **2.24** | 3.29 | 4.35 | **2.24** |

## 5.2 Superior testing accuracy analysis: from training optimality perspective

To figure out the reason behind the superior testing accuracy, we then analyze the optimality of training. Training trees with different depths correspond to different optimization tasks. To assess optimality, training trees at fixed depth is a common practice. Fixed depths of $D = \{2, 4, 8, 12\}$ are used for comparison. Table 3 shows that our RADDT outperforms ORT-LS by 5.35%, 2.66% and 0.02% in training accuracy for depths of 2, 4, and 8, respectively, while it outperforms CART by 24.83%, 21.02% and 9.48% across various depths. For other gradient-based trees, RADDT significantly outperforms GradTree and SoftDT on average at all depths. LatentTree and DGT do not scale to depth 12. On average across these feasible depths, our method outperforms LatentTree by 10.39% in training and 6.24% in testing accuracy, and surpasses DGT by 7.52% in training and 2.66% in testing accuracy. Improvements in training accuracy imply that our optimization algorithm achieves better training optimality. Furthermore, an increase in training accuracy correlates with the improved test accuracy at depths of 2, 4, and 8, suggesting that an optimized tree with higher training accuracy can potentially yield better test accuracy before encountering serious overfitting issues. Overfitting, particularly at a depth of 12, is simply addressed by tuning an optimal tree depth. The overfitting limitation is further analyzed in Section 5.6.

Table 3: Fixed-depth comparison of training, testing accuracy and training time on Group (i) datasets.

|  | D | Greedy Methods | | | | Gradient-based Trees | | | | Local Search | Ours |
|---|---|---|---|---|---|---|---|---|---|---|---|
|  |  | CART | RankCART | HHCART | OC1 | GradTree | SoftDT | DGT | LatenTree | ORT-LS | RADDT |
| Train ($R^2$,%) | 2 | 46.95 | 32.81 | 46.20 | 49.59 | 39.42 | 50.59 | 64.48 | 66.94 | 66.43 | **71.78** |
|  | 4 | 61.16 | 54.09 | 62.65 | 63.21 | 53.29 | 56.83 | 75.83 | 72.25 | 79.52 | **82.18** |
|  | 8 | 81.45 | 77.51 | 82.26 | 81.43 | 64.65 | 66.81 | 82.01 | 74.54 | 90.91 | **90.93** |
|  | 12 | 93.45 | 93.06 | 94.74 | 93.02 | 66.86 | 73.03 | / | / | 97.46 | 96.36 |
| Test ($R^2$,%) | 2 | 46.14 | 32.47 | 45.61 | 47.95 | 38.53 | 49.94 | 63.81 | 66.42 | 64.51 | **70.07** |
|  | 4 | 58.92 | 52.69 | 61.30 | 60.22 | 51.45 | 56.32 | 75.16 | 71.03 | 75.06 | **78.98** |
|  | 8 | 69.77 | 69.93 | 74.57 | 69.08 | 62.40 | 66.44 | 79.99 | 70.77 | 74.93 | **77.89** |
|  | 12 | 68.28 | 64.13 | 68.37 | 65.57 | 63.81 | 72.45 | / | / | 68.25 | **73.11** |
| Time (s) | 2 | 0.03 | 0.70 | 4.43 | 3,216 | 31.32 | 22.24 | 2,102 | 1,624 | 457.21 | 542.22 |
|  | 4 | 0.04 | 1.56 | 7.45 | 4,192 | 54.74 | 81.88 | 2,577 | 2,194 | 868.08 | 478.81 |
|  | 8 | 0.07 | 5.08 | 12.52 | 4,803 | 298.92 | 1,209 | 4,049 | 2,381 | 9,336 | 602.69 |
|  | 12 | 0.10 | 12.61 | 22.01 | 5,103 | 10,417 | 54,829 | / | / | 210,141 | 2,643 |

**Training optimality comparison against global optimal solutions on Group (iii) datasets:** The above results reveal that our approach achieves superior training optimality. To provide a reference for global optimality, we then compare it with global optimal trees ORT-MIP. However, ORT-MIP fails to find optimal trees for any of the above datasets, even with 128 cores and a two-day time limit. It can only solve a 2-depth tree on 4 small datasets in Group (iii) to a global optimum with 1% optimality gap. Despite the existence of sparse optimal trees [Zhang et al., 2023], a direct comparison is infeasible for some reasons discussed in Appendix K.7. We also detail train accuracy comparison there. Results show that RADDT nearly reaches global optimum (ORT-MIP) with 2.82% discrepancy.

**Training optimality comparison against ground truth on Group (iii) datasets:** We further evaluate RADDT on three synthetic datasets with known ground truth, each with 5000 samples. The procedure for generating synthetic datasets is detailed in Appendix H. The corresponding ground truths for training and testing accuracy are 100%. In Appendix K.8, Table 20, our RADDT closely approximates global optimality in training and testing accuracy, with 0.64% difference over ground truth on average. These results, together with the previous global solution comparisons, further validate the effectiveness of our tree optimization.

**Ablation study on the optimization strategies:** The notable training accuracy of our RADDT method can be attributed to two key strategies designed to enhance approximation accuracy: the multi-run warm start annealing strategy, which improves training accuracy by an average of 7.1% across various depths compared to standard softmin function without any scaling; and the strategy of adjusting

initial solutions for gradient-based optimization, which contributes an additional 1.8% improvement on average. Detailed results for the ablation study are provided in Appendix K.9.

## 5.3 Scalability assessment to 12-depth trees on 7 million-scale datasets in Group (iv)

Previous experiments focus on three different groups of datasets with fewer than 41k samples. To further evaluate our scalability, we test it on seven large-scale datasets, each containing at least 1 million samples, using the same fixed-depth setting as in Table 3. The Group (iv) dataset is detailed in Appendix H. As shown in Appendix K.10, Table 22, our `RADDT` successfully scales to a 12-depth tree, outperforming `CART` by 4.83% in training and 3.15% in testing across all depths. In contrast, `GradTree` and `SoftDT` fail to output good solutions. At the feasible depths for `DGT` and `LatentTree`, `RADDT` outperforms `DGT` by 3.9% in training and 3.74% in testing on average, while it slightly underperforms `LatentTree` by 0.03% in training and 0.08% in testing. However, `RADDT` takes significantly less training time, and `LatentTree` is only solvable at depths 2 and 4. These comparisons effectively showcase `RADDT`'s scalability on million-scale datasets at depth-12 trees.

**GPU acceleration and training time analysis:** Our training time is primarily affected by sample size $n$, feature number $p$ and depth $D$. For $D$-depth tree training, each sample must be evaluated at all $2^D - 1$ branch nodes via matrix operation $\boldsymbol{A}\boldsymbol{x}_i$, requiring $\mathcal{O}(p \cdot (2^D - 1))$ operations per sample and $\mathcal{O}(n \cdot p \cdot (2^D - 1))$ operations for the full dataset. It indicates that training time increases with larger $n$ and $D$. Importantly, our implementation is highly parallelizable and facilitates multi-GPU acceleration. The specific GPU configurations in our experiments and acceleration analysis are given in Appendix K.11. In the million-scale comparison Appendix K.10, Table 22, our `RADDT` achieves a 42-fold speedup over `DGT` at depth 4. These scalability advantages become even more pronounced with deeper trees and more samples, where larger matrix operations dominate the computation. However, in this setting within Table 22, most high-accuracy baselines fail to scale to depth 12, making direct comparisons at that depth infeasible. To further illustrate the advantage, we refer to Table 3, where the second-best method `ORT-LS` scales to depth 12. For instance, on the "ailerons" dataset, `RADDT` is 432 times faster than `ORT-LS`, demonstrating the substantial speedup at deeper depths. Detailed time comparisons for datasets with more than 10,000 samples used in Table 3 are given in Appendix K.12. Besides, a training time comparison of our method in GPU versus CPU settings is provided in Appendix K.13, although the advantages of GPU acceleration are expected.

## 5.4 Analysis of model complexity and inference time

Alongside tree depth, the number of model parameters also offers a clear view of model complexity. While oblique tree typically requires more parameters than orthogonal tree at a given depth due to feature combinations, it often yields better accuracy with much smaller depth. Therefore, for a similar accuracy, our oblique tree may not require more parameters overall than orthogonal tree.

For a $D$-depth binary tree with $\mathbb{T}_B = 2^D - 1$ branch nodes and $\mathbb{T}_L = 2^D$ leaf nodes, and a dataset with $p$ features, orthogonal tree like CART requires $2 \cdot \mathbb{T}_B + \mathbb{T}_L$ parameters. In contrast, our oblique tree `RADDT` requires $\mathbb{T}_B \cdot p + 1 + \mathbb{T}_L$ parameters, while our `RADDT-Linear` needs $\mathbb{T}_B \cdot p + 1 + \mathbb{T}_L \cdot p + 1$ parameters. Based on these, we compare the number of model parameters for `CART`, `RADDT` and `RADDT-Linear` in Table 4 by averaging across all Group (i) 17 datasets. As shown in Table 4, `CART` achieves 74.85% accuracy with average depth 10.24 and 3,140.94 parameters. Our `RADDT` and `RADDT-Linear` reach 82.39% and 84.63% accuracy at depths 7.29 and 4.82, using 10,503.94 and 4,044.71 parameters, respectively. Notably, if the goal is to match or slightly exceed `CART`'s accuracy, our models can do so with much smaller trees. Table 4 shows that `RADDT` achieves 75.23% at depth 2.82 using only 101.71 parameters, and `RADDT-Linear` reaches 77.21% at depth 1 with just 40.47 parameters, yielding 31 times and 78 times reductions in parameter count compared to `CART`. These results indicate that our methods do not necessarily require more parameters for improved accuracy; the total parameter cost depends on the desired trade-off between accuracy and model complexity.

Additionally, inference time greatly helps assess model complexity. In the setting of Table 1 and Table 2, our methods achieves superior accuracy improvements but involves more parameters than `CART`, resulting in slightly higher inference time. However, if only aiming to achieve an accuracy comparable to `CART` (74.85%), Table 4 shows that the inference time of `RADDT` becomes comparable to `CART`, while `RADDT-Linear` is even faster. It is important to note that our implements use Python

Table 4: Model complexity and inference time comparison with `CART` on Group (i) 17 datasets.

| Method | Testing Accuracy ($R^2$, %) | Depth | Number of Branch Nodes | Number of Leaf Nodes | Total Number of Model Parameters | Inference Time (Millisecond) |
|---|---|---|---|---|---|---|
| `CART` | 74.85 (reported in Table 1) | 10.24 | 1,046.65 | 1,047.65 | 3,140.94 | 0.44 |
| `RADDT` | 82.39 (reported in Table 3) | 7.29 | 671.94 | 672.94 | 10,503.94 | 1.88 |
| | 75.23 (only slightly exceed CART's accuracy) | 2.82 | 6.88 | 7.88 | 101.71 | 0.54 |
| `RADDT-Linear` | 84.63 (reported in Table 3) | 4.82 | 129 | 130 | 4,044.71 | 1.61 |
| | 77.21 (only slightly exceed CART's accuracy) | 1 | 1 | 2 | 40.47 | 0.39 |

and PyTorch, where inference time may not scale strictly with parameter count due to framework overhead. A pure C++ implementation would likely align more closely with parameter count.

## 5.5 Interpretability analysis

We fully acknowledge that orthogonal trees like `CART`, at the same depth, are generally more interpretable than oblique trees. However, as originally analyzed in `OC1` [Murthy et al., 1994], oblique trees could become more interpretable when their depth is substantially smaller. Interpretability depends on multiple factors including tree depth, feature number, and the accuracy-simplicity trade-off. There is no interpretability superiority of one approach over the other, as it depends on specific application scenarios. We evaluate our `RADDT`'s interpretability from three aspects: tree-based prediction logic, unique prediction path and the complexity of decision rules. Detailed analysis can be found in Appendix K.14 with an illustrative example in Figure 7.

## 5.6 Limitations analysis and future work

While our approach achieves superior accuracy, several limitations remain. The primary limitation lies in inadequate regularization. In Table 3, while training accuracy for `RADDT` improves by 5.43% from depth 8 to 12, testing accuracy conversely drops by 4.78%, indicating a serious overfitting issue. This issue is more pronounced in trees with linear predictions, as detailed in Appendix K.15. In response, we apply preliminary $L1$ regularization to `RADDT-Linear` for experiments in Table 2, leading to a modest 0.57% improvement in testing accuracy. Due to the challenges in identifying optimal regularization strength and potential increases in training time, we just used a small value of $1e-4$, without extensive tuning. Despite this, further improvements could be achieved through more dedicated regularization strategies, such as the post-hoc regularization technique, Hierarchical Shrinkage, proposed for trees by Agarwal et al. [2022]. Second, the effectiveness of our multi-run warm start annealing strategy is validated by empirical observations; however, a theoretical analysis is still lacking. Third, our tailored initialization adjustment strategy fails to consider the corner case of an ill-defined initialization where both $\boldsymbol{a}_j = 0$ and $b_j = 0$. This is also a limitation, despite its rarity and minimal impact on empirical performance in practice. Fourth, while our current hyperparameter tuning for tree ensembles is sufficient to support our main claims, especially given that our goal is not to claim superiority over ensembles, more tuning on regularization parameters could potentially yield even better results for tree ensembles. Exploring this gap between a single tree and ensembles presents a promising area for future research. Next interesting future direction is to apply our optimization strategies, such as initialization and multi-run warm-start annealing, to other gradient-based trees like `SoftDT`. Since these strategies are general, they are not limited to our specific tree formulation and may benefit other models. Another promising alternative to our scaled softmin approximation is the Entmax function. However, its application requires further investigation, as selecting its sparsity parameter presents a similar challenge to choosing the scale factor in our softmin method.

## 6 Conclusion

Our approach addresses two main limitations of classic decision tree: non-differentiability and low test accuracy, thereby enhancing its practicality. We tackle these issues by first proposing a "ReLU+Argmin"-based hard split tree with a novel exact reformulation, then significantly improving its test accuracy through our gradient-based entire tree optimization. The entire tree training problem is reformulated as an unconstrained optimization task, with a scaled softmin function approximating Argmin behaviors for effective gradient backpropagation. Our reformulated tree, unlike existing soft approximation, eliminates the softness at both branch nodes and leaf nodes, thus preserving hard splits and deterministic predictions. Extensive experiments demonstrate our tree's superior accuracy and scalability on million-scale datasets.

## Acknowledgments and Disclosure of Funding

Yankai Cao acknowledges funding from discovery program of the Natural Science and Engineering Research Council of Canada (RGPIN-2019-05499) and New Frontiers in Research Fund (NFRFE-2022-00663). The authors also gratefully acknowledge the computing resources and services provided by Digital Research Alliance of Canada (www.alliancecan.ca), and Advanced Research Computing at the University of British Columbia.

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

## Appendices for:
## Differentiable Decision Tree via "ReLU+Argmin" Reformulation

## A  Certain Scenarios Requiring Decision Trees with Hard-splits

Hard-split decision trees and soft decision trees represent fundamentally different models, each suited to different application scenarios. Further, there do exist scenarios where the application of hard-split decision trees is not only more appropriate but also imperative. Below is an illustrative example from industrial projects that underscores this preference.

Scenario: Piece-wise affine control law in explicit model predictive control.

In explicit model predictive control, the control law is often represented by piece-wise affine functions, which can be effectively approximated by a hard-split decision tree. The hard decisions at branch nodes can be used to determine the control action based on the state of the system, which can be easily implemented in real-time control systems.

Beyond the utility of making clear hard decisions, the optimal decision tree model that offers superior predictive performance is also crucial for these scenarios. From the perspective of application scenarios, the motivation and necessity for hard-split optimal decision trees are intuitive and compelling.

## B  Empirical Analysis of Scale Factor Impacts on Scaled Softmin Operation

The scale factor $\alpha$ significantly influences the approximation degree to Argmin function and the behavior of its gradient in optimization. A larger $\alpha$ leads to a better approximation than standard softmin function with $\alpha = 1$, achieving closer proximity to an Argmin function as $\alpha$ approaches infinity. An illustration for the gap between scaled softmin function and the Argmin function under varying $\alpha$ is shown in Figure 6. Nonetheless, a larger $\alpha$ also results in a more unstable gradient, which may adversely affect the optimization process. Consequently, the $\alpha$ represents a critical balance between achieving high approximation degree and maintaining stability in optimization processes.

To show how the scale factor $\alpha$ impacts the training optimality of the unconstrained optimization problem (our RADDT method without using the strategy of multi-run warm start annealing), we conduct comparison experiments across different $\alpha$ values at different depths. Our differentiable decision tree optimization framework will be introduced in Section 4. The findings, summarized in Table 5, reveal the relationship between $\alpha$ and training performance, as measured by the average training accuracy $R^2$. Notably, we observe that the training accuracy at the extremes of $\alpha = 1$ (standard softmin function) and $\alpha = 1000$ are inferior compared to intermediate $\alpha$ values. This observation underscores two critical insights: first, relying solely on softmin function ($\alpha = 1$) yields suboptimal optimization results; second, the high $\alpha$ value may not necessarily lead to better optimality.

Table 5: The impact of $\alpha$ on training accuracy across different depths.

| Various $\alpha$ Value | Training Accuracy (%) | | | |
|---|---|---|---|---|
| | $D = 2$ | $D = 4$ | $D = 8$ | $D = 12$ |
| $\alpha = 1$ (Standard Softmin Function) | 60.78 | 69.97 | 81.46 | 93.45 |
| $\alpha = 20$ | 70.28 | 79.43 | 87.98 | 94.49 |
| $\alpha = 150$ | 68.00 | 78.31 | 88.08 | 94.42 |
| $\alpha = 1000$ | 65.05 | 74.16 | 83.79 | 93.45 |

However, it remains a challenge to identify the optimal scale factor $\alpha$ that balances the trade-off between approximation degree and differentiability. To mitigate this issue, we propose a multi-run warm start annealing strategy, detailed in the following Section 4.1, to narrow the gap between the Argmin behaviors and its differentiable approximation.

## C  Deterministic Calculations for Leaf Prediction Parameters

As detailed in Section 3.2, our method deterministically calculates the leaf prediction parameters $\theta$ (i.e., the parameters $K$ and $h$ in regression, and the parameters $h$ in classification), instead of directly using trained values for $\theta$. Given a tree with tree split parameters $A$ and $b$, the deterministic tree path

for each sample can be obtained, which allows determining the total number of samples assigned to a specific leaf node $t \in \mathbb{T}_L$.

In classification tasks, the value of $\boldsymbol{h}$ at a leaf node $t$ corresponds to the majority class among the samples assigned to that leaf node. In this case, the leaf value is represented as a class label or the one-hot vector specific to the class, $h_t \in \{0, 1\}^c$, in contrast to the continuous value $h_t \in R^c$ used during the training phase.

In regression tasks, we consider two types of leaf prediction: linear prediction and constant prediction. For decision trees with linear predictions, the prediction at a leaf node is a linear combination of input features by fitting a linear correlation between all samples assigned to that leaf node. The leaf values at a leaf node $t$, $\boldsymbol{k}_t$ and $h_t$, are the linear coefficients determined by linear regression. For decision trees with constant predictions, the value of $\boldsymbol{K}$ remains zero. The value of $\boldsymbol{h}$ at a leaf node $t$ is an average of the true output values ($y_i$) of the samples assigned to that leaf node $t$.

# D  Implementation Details of Multi-run Warm Start Annealing

As discussed in Section 4.1, the key challenge in enhancing the approximation accuracy to Argmin behaviors lies in the selection of $\alpha$. A larger $\alpha$ may destabilize optimization process, whereas a smaller $\alpha$ tends to be easier to solve for gradient-based optimization. The annealing strategy, which gradually changes $\alpha$ from small to large values, has been ever used to strike such a balance in existing literature [Hehn and Hamprecht, 2017, Karthikeyan et al., 2022, Lee and Jaakkola, 2020], though not specifically for our application to scaled softmin. However, their annealing typically adjusts $\alpha$ within a single training process, such as changing $\alpha$ every few epochs, which may affect stable gradient updates. Besides, the implementation details in these works are not well-documented. Considering these, we introduce a detailed description of our annealing strategy.

Our annealing process operates across multiple optimization runs with different $\alpha$ values. The solution from an optimization task with a smaller $\alpha$ is used to effectively warm-start the next optimization run with a larger $\alpha$. By starting with a smaller scale factor and gradually increasing it, this strategy enhances the approximation accuracy, while mitigating numerical instability typically associated with larger scale factors.

Specifically, the procedure begins by sampling a set of scale factors within a predetermined range from the smallest $\alpha_{min}$ to the largest $\alpha_{max}$, ensuring a broad exploration of possible $\alpha$ values. We adopt log-space sampling to generate these scale factors, $\{\alpha_1, \cdots, \alpha_n\}$, evenly on a logarithmic scale. In this case, these sampled scale factors can be denser in the smaller range, which is beneficial for the stable optimization process. We initiate the optimization with the smallest sampled scale factor to generate the initial optimized tree candidate. This candidate then serves as the starting point for the next optimization run with a slightly larger $\alpha$. This iterative process is repeated until all sampled scale factors have been utilized. Detailed implementation steps are integrated within our systematic optimization framework, Algorithm 1.

## D.1  Comparison between our annealing strategy and binary search for scale factor selections

Binary search provides a structured way to explore different $\alpha$ values for scaling the softmin approximation. Unlike random trials, it narrows the search range by evaluating performance at the midpoint and adjusting boundaries accordingly. While this is more principled than random guessing, it still involves testing isolated $\alpha$ values. As earlier discussed, this approach does not match the performance of our multi-run warm start annealing strategy.

While we agree that a binary search-like approach can be used to identify an effective $\alpha$, our empirical results show that gradually increasing $\alpha$, rather than directly using a large fixed value, leads to better training outcomes. This is because starting with a smaller $\alpha$ allows the model to optimize under a smoother case, improving stability. As $\alpha$ increases, it becomes sharper and closer to the ideal formulation, and the model benefits from being progressively adapted to this more accurate approximation. Our annealing strategy takes advantage of this effect by warm-starting each successive optimization task with the solution from previous task with smaller $\alpha$. This gradual refinement enhances the approximation accuracy, while mitigating numerical instability typically associated with larger $\alpha$.

The average training accuracy across Group **(i)** 17 regression datasets for different tree depths is summarized below. Our annealing strategy consistently outperforms binary search, with an average improvement of 2.82% across all depths.

Table 6: Training accuracy (optimization capability) comparison with different $\alpha$ selection strategies.

| Tree Depth | 2 | 4 | 8 | 12 |
|---|---|---|---|---|
| Using Binary Search for $\alpha$ Selections | 69.95 | 77.33 | 87.69 | 94.98 |
| Using Our Multi-run Warm Start Annealing Strategy for $\alpha$ Selections | 71.78 | 82.18 | 90.93 | 96.36 |

These results validate the effectiveness of our annealing in practice. However, we acknowledge that this finding is based on empirical observation, and a theoretical analysis is still lacking.

# E  An Illustrative Example of The Influence of Initial Solutions and Their Adjustments for Gradient-based Optimization

As discussed in Section 4.2, a good initial solution is crucial for achieving a better optimality, especially in our task with softmin operations. This is because our softmin operates on $U_{i,t}$, which is ReLU-based violations with either zero or non-zero values. However, these non-zero violations may be close to zero, especially for samples around the decision boundary $\boldsymbol{a}_j^T \boldsymbol{x}_i - b_j = 0$. While such small discrepancies between exact-zero and near-zero violation do not impact Argmin behaviors, which select the unique path only when $U_{i,t} = 0$, they can sensitively affect softmin. Specifically, softmin may fail to produce values close to one as ideally expected, potentially compromising approximation accuracy. Here, we provide an illustrative example to understand the influence of initial solutions and their adjustments for gradient-based optimization. As shown in Figure 3, the red sample $x^*$ lies near the blue decision boundary, resulting in a small positive ReLU output, close to zero, that is difficult to distinguish from an exact zero.

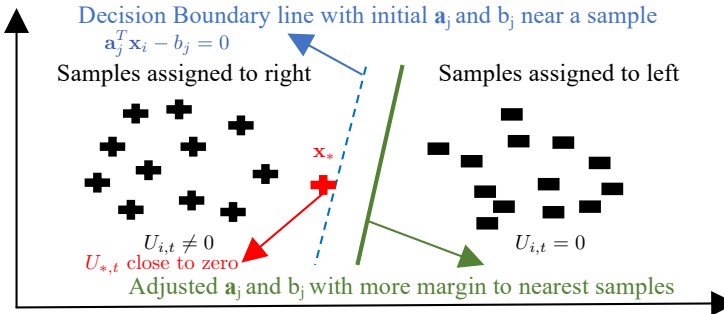

Figure 3: Illustration of the influence of initial solutions and the effective adjustments.

To mitigate this, the initial $\boldsymbol{A}$ and $\boldsymbol{b}$ can be adjusted so that the decision boundary achieves the maximum margin from the nearest samples, without altering the existing sample assignments as illustrated by the green line in Appendix E, Figure 3. This adjustment does not change sample assignments but enhances the distinction between exact-zero and near-zero cases.

The adjustment can be easily implemented either by calculating the median of $\boldsymbol{a}_j^T \boldsymbol{x}_i$ for the closest samples on both sides of the boundary, or using support vector machine to treat sample assignments as a binary classification task. To be more specific, Each branch test $\boldsymbol{a}_t^T \boldsymbol{x}_i \leq b_t$ acts as a binary classifier, directing samples left or right. A practical adjustment is to slightly shift the decision boundary (e.g., by modifying the median of $\boldsymbol{a}_j^T \boldsymbol{x}_i$) to enlarge the margin without changing sample assignments. Beyond only adjusting $b_j$, we also refine both $\boldsymbol{a}_j$ and $b_j$ using an SVM: we treat left/right assignments as binary labels and fit a linear SVM to maximize margin. The resulting weights and bias update the split parameters while preserving the original assignment. This procedure is visualized in Appendix E, Figure 3.

# F  Hyperparameters Analysis of Our Differentiable Decision Tree Optimization Approach

Despite the introduction of additional hyperparameters in gradient-based optimization, tuning them is not typically necessary because their effects are straightforward. To be more specific, the hyperparameters in our differentiable decision tree optimization approach are as follows:

(1) The multi-start number $N_{start}$

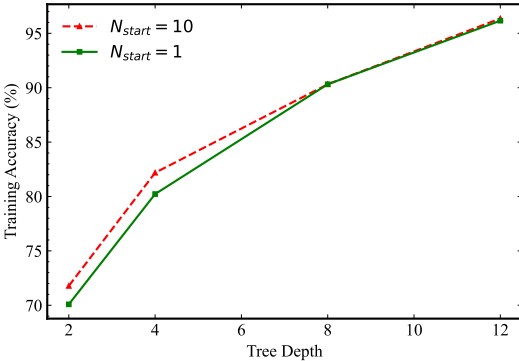

Figure 4: The trend of training optimality under different depth setting with different $N_{start}$.

The multi-start number $N_{start}$ directly influences training optimality by increasing the chance of finding the optimal solution, albeit at a higher computational cost. In practice, $N_{start}$ is set to balance acceptable computational expenses with desired training accuracy.

To explore the correlation between $N_{start}$ and the training optimality, our "ReLU+Argmin"-based Differentiable Decision Tree Optimization (RADDT) for regression trees with constant predictions is performed under different $N_{start}$ values as shown in Figure 4. It indicates that increasing $N_{start}$ generally improves training optimality for all various tree depths, especially at lower tree depths.

(2) The epoch number $N_{epoch}$

The epoch number $N_{epoch}$ is another hyperparameter that directly affects training optimality. A higher $N_{epoch}$ value increases training accuracy, but it also increases computational costs. In practice, $N_{epoch}$ is also set to balance acceptable computational expenses with desired training accuracy.

Our experiment with different $N_{epoch} = \{100, 3000, 5000\}$ in Figure 5, shows that increasing $N_{epoch}$ generally improves training optimality for all various tree depths.

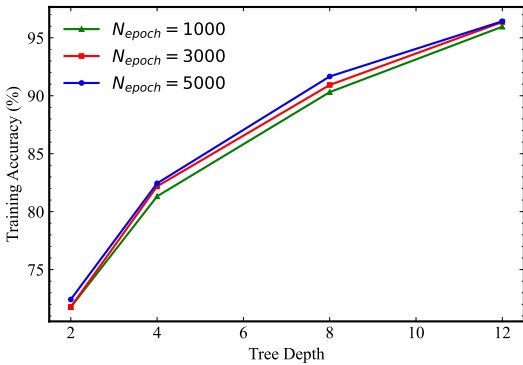

Figure 5: The trend of training optimality under different depth setting with various $N_{epoch}$.

(3) The range of sampled scale factors $[\alpha_{min}, \alpha_{max}]$ for scaled softmin function

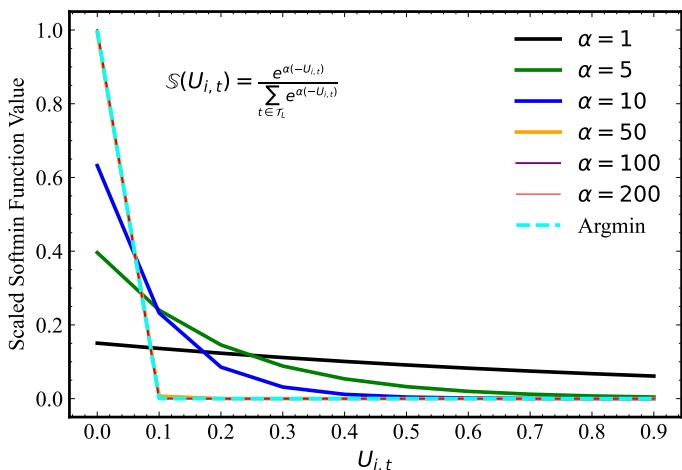

Figure 6: Comparison of scaled softmin function approximating Argmin behaviors under varying $\alpha$.

This predetermined range is used to sample a set of scaled factors $\alpha$ for the strategy of multi-run warm start annealing. The principal aim is to explore a broader range of scale factors, ranging from smaller to larger values. As discussed in Section 3.2, the scaled softmin function applied to $U_{i,t}$ is used to approximate the Argmin function (more concretely, the one-hot encoded Argmin function). A larger $\alpha$ leads to a closer approximation to the Argmin function. As observed in Figure 6, the gap between scaled softmin function and the Argmin function narrows as $\alpha$ increases. Noticeably, the standard softmin function with $\alpha = 1$ exhibits a significant deviation from the Argmin function as depicted in the black line. With a larger $\alpha$, such as $\alpha \geq 50$, the scaled softmin function closely approximates the Argmin function, almost overlapping with the Argmin function as shown in the figure. In the implementation of our experiments, we simply sample $\alpha$ within the range $[\alpha_{min}, \alpha_{max}] = [2, 200]$ to meet our requirements. This range ensures that smaller values maintain a smooth gradient and exhibit a better approximation than the standard softmin function, while larger values closely approximate the Argmin function.

(4) The number of sampled scale factors

Within a predetermined range, a set of scaled factors, denoted as $\{\alpha_1, \cdots, \alpha_n\}$, are sampled for subsequent use in multi-run warm start annealing, as detailed in Section 4.1. Larger scale factors reduce the approximation disparity, whereas smaller ones yield a smoother and more stable gradient. Including a greater number of scale factors in the set facilitates a more stable approximation process, enhancing the approximation degree to Argmin behaviors while minimizing the loss of differentiability typically associated with larger scale factors. Intuitively, including more scale factors in the set enhances training optimality. However, this leads to increased iterations in the annealing strategy, thereby raising computational costs. Practically, the number of scale factors is often determined by balancing training accuracy against computational demands. In our experiments, to avoid excessive computational costs, we primarily sample 5 scale factors with a log-space distribution within the range $[\alpha_{min}, \alpha_{max}]$.

(5) The learning rate $\eta$

The learning rate is a common parameter in gradient-based optimization, and has garnered significant attention in the literature. To simplify its usage, we adopt the well-established learning rate scheduler, termed `CosineAnnealingWarmRestarts` (with initial linear warm up) in PyTorch, which decreases the learning rate from an initial value of 0.01, thus minimizing the need for additional tuning.

## G   Algorithm Details of "ReLU+Argmin"-based Differentiable Decision Tree

Alongside the detailed descriptions in Section 4.3, we present the algorithmic implementation of our "ReLU+Argmin"-based Differentiable Decision Tree Optimization framework, `RADDT`, in Algorithm 1.

---
**Algorithm 1** The entire tree optimization framework for `RADDT`.
---
1: **Input:** $\{\boldsymbol{x}_i, y_i\}_{i=1}^n$, tree depth $D$, learning rate $\eta$, epoch number $N_{epoch}$, multi-start number $N_{start}$.
2: **Output:** Optimal trainable variables $\boldsymbol{A}_{best}$, $\boldsymbol{b}_{best}$, $\theta_{best}$.
3: Define $\mathcal{L}_{min}$, empty variables $\boldsymbol{A}_{best}$, $\boldsymbol{b}_{best}$ and $\theta_{best}$.
4: **for** $start = 1$ **to** $N_{start}$ **do**
5:  Initialize $\boldsymbol{A}$, $\boldsymbol{b}$, $\theta$ with initial solution adjustments as detailed in Section 4.2. Generate scale factors $\{\alpha_1, \cdots, \alpha_n\}$ with log-spacing.
6:  **for** $\alpha_{iter} \in \{\alpha_1, \cdots, \alpha_n\}$ **do**
7:    If $iter \neq 1$, initialize trainable variables with the solution of last optimization run by *Line 12*.
8:    **for** $k = 1$ **to** $N_{epoch}$ **do**
9:      Calculate loss $\mathcal{L}$ at step $k$ by Equation (7) and calculate $\frac{\partial \mathcal{L}}{\partial \boldsymbol{A}}$, $\frac{\partial \mathcal{L}}{\partial \boldsymbol{b}}$, $\frac{\partial \mathcal{L}}{\partial \theta}$. Then update trainable variables via gradient decent, such as $\boldsymbol{A}_{k+1} = \boldsymbol{A}_k - \eta \frac{\partial \mathcal{L}}{\partial \boldsymbol{A}}$.
10:    **end for**
11:    Deterministically update $\theta$ without any approximation, as described in Appendix C.
12:    Generate a tree candidate with optimized variables, termed as $\boldsymbol{A}_{iter}$, $\boldsymbol{b}_{iter}$, $\theta_{iter}$. Specifically, $\boldsymbol{A}_{iter} = \boldsymbol{A}_{N_{epoch}}$ and $\boldsymbol{b}_{iter} = \boldsymbol{b}_{N_{epoch}}$, while $\theta_{iter}$ is derived through a deterministic recalculation as mentioned in *Line 11*.
13:    Deterministically compute $\mathcal{L}$ by Equation (5). IF $\mathcal{L} < \mathcal{L}_{min}$, update variables like $\boldsymbol{A}_{best} \leftarrow \boldsymbol{A}_{iter}$.
14:  **end for**
15: **end for**
---

## H  Detailed Description and Usage of The Four Groups of Datasets

### Dataset split ratio and usage

In our experiments, typically, we allocate 75% of the samples for training purposes and the remaining 25% for testing, following the train-test split ratio as used in [Bertsimas and Dunn, 2017]. If an experiment requires cross validation for hyperparameter tuning like tree depth, we then subdivide the training datasets into training and validation subsets in a 2:1 ratio. The dataset setting accordingly changes to 50% samples as training set, 25% samples as validation set, and 25% samples as testing set. After determining the best hyperparameters, we then retrain the model using the combined training and validation set, and use the remaining 25% as the testing set to evaluate the final testing accuracy.

### Dataset information across four groups

As discussed in Section 5, our baseline comparison experiments are conducted across four groups of datasets: **(i)** 17 medium-sized regression datasets with fewer than 41k samples (primary focus); **(ii)** 27 same classification and 9 regression datasets used in certain baselines (for more credible comparison on their used datasets); **(iii)** 4 real-world datasets with about 100 samples, and 3 synthetic datasets with 5000 samples (for comparing global optimality with global tree `ORT-MIP` [Bertsimas and Dunn, 2017] and known ground truth, respectively); **(iv)** 7 large-scale datasets with at least 1 million samples (for scalability).

Unless otherwise specified, these real-world datasets used for our regression experiments are collected from the UCI repository [Dua and Graff, 2019] and OpenML [Vanschoren et al., 2014]. Regarding the 17 medium-sized datasets in Group **(i)**, detailed information about these datasets is summarized in Table 7. The dataset size $n$ and the number of features $p$ are provided in the table.

Regarding the same classification datasets used in the original `GradTree` study [Karthikeyan et al., 2022] in Group **(ii)**, we include these shared datasets to support a more credible and convincing comparison. Although we have already compared against `GradTree` on the Group **(i)** 17 regression datasets in Table 7, and their subpar performance aligns with the results reported in their own Table 1 and 2, where their method underperforms the foundational baseline `CART` on 17 out of 36 classification datasets, we still further evaluate on the same classification datasets used in their own work to provide a more credible comparison. Among their 36 classification datasets, we finally collect 27 and discard 9 datasets, due to one or more of the following issues: unavailable dataset sources, unusable datasets with substantial missing values (corrupted dataset), or ambiguous dataset documentation. For example, the "rice" dataset lacks a clear download source. The "annealing" dataset contains a significantly large number of missing values and is nearly unusable. The "splice" dataset lacks clear information, such as the definition of features and predictive target, making this type of dataset unusable. Therefore, to ensure experimental reproducibility and data integrity, we

Table 7: Group (i) 17 real-world regression datasets from UCI and OpenML Repository.

| Dataset Index | Dataset Name | Dataset Size (n) | Feature Number (p) |
|---|---|---|---|
| 1 | airfoil-self-noise | 1,503 | 5 |
| 2 | space-ga | 3,107 | 6 |
| 3 | abalone | 4,177 | 8 |
| 4 | gas-turbine-co-emission-2015 | 7,384 | 9 |
| 5 | gas-turbine-nox-emission-2015 | 7,384 | 9 |
| 6 | puma8NH | 8,192 | 8 |
| 7 | cpu-act | 8,192 | 21 |
| 8 | cpu-small | 8,192 | 12 |
| 9 | kin8nm | 8,192 | 8 |
| 10 | delta-elevators | 9,517 | 6 |
| 11 | combined-cycle-power-plant | 9,568 | 4 |
| 12 | electrical-grid-stability | 10,000 | 12 |
| 13 | condition-based-maintenance_compressor | 11,934 | 16 |
| 14 | condition-based-maintenance_turbine | 11,934 | 16 |
| 15 | ailerons | 13,750 | 40 |
| 16 | elevators | 16,599 | 18 |
| 17 | friedman-artificial | 40,768 | 10 |

discard 9 datasets in total and perform the same-dataset comparison on the remaining 27 datasets. Additionally, since their work adopts a 20% test split, different from our default setting, we adopt their train-test split ratio for this specific comparison to ensure fairness. The final list of datasets, along with their sample size $n$, number of features $p$, and number of classes $c$, is presented in Table 8.

Table 8: Group (ii) 27 same classification datasets used in the original `GradTree` study.

| Dataset Index | Dataset Name | Dataset Size (n) | Feature Number (p) | Class Number (c) |
|---|---|---|---|---|
| GradTree-1 | balance-scale | 625 | 4 | 3 |
| GradTree-2 | banknote-authentication | 1,372 | 4 | 2 |
| GradTree-3 | blood-transfusion | 748 | 4 | 2 |
| GradTree-4 | car-evaluation | 1,728 | 6 | 4 |
| GradTree-5 | congressional-voting-records | 232 | 16 | 2 |
| GradTree-6 | contraceptive-method-choice | 1,473 | 9 | 3 |
| GradTree-7 | dermatology | 358 | 34 | 6 |
| GradTree-8 | echocardiogram | 61 | 11 | 2 |
| GradTree-9 | iris | 150 | 4 | 3 |
| GradTree-10 | spambase | 4,601 | 57 | 2 |
| GradTree-11 | thyroid-disease-ann-thyroid | 7,200 | 21 | 3 |
| GradTree-12 | wine | 178 | 13 | 3 |
| GradTree-13 | adult | 32,561 | 14 | 2 |
| GradTree-14 | bank-marketing | 45,211 | 14 | 2 |
| GradTree-15 | credit-card | 30,000 | 23 | 2 |
| GradTree-16 | german | 1,000 | 20 | 2 |
| GradTree-17 | glass | 214 | 9 | 6 |
| GradTree-18 | heart-failure | 299 | 12 | 2 |
| GradTree-19 | landsat | 6,435 | 36 | 6 |
| GradTree-20 | loan-house | 614 | 11 | 2 |
| GradTree-21 | lymphography | 148 | 18 | 4 |
| GradTree-22 | mushrooms | 8,124 | 22 | 2 |
| GradTree-23 | raisins | 900 | 7 | 2 |
| GradTree-24 | segment | 2,310 | 19 | 7 |
| GradTree-25 | solar-flare | 1,389 | 10 | 8 |
| GradTree-26 | wisconsin-breast-cancer | 569 | 10 | 2 |
| GradTree-27 | zoo | 101 | 16 | 7 |

Regarding the same regression datasets used in the work of soft decision trees, no tabular datasets were used in [Frosst and Hinton, 2017], but we adopt the four tabular regression datasets from the original soft decision tree study by [İrsoy et al., 2012]. We include these shared datasets to support a more credible and convincing comparison as well. The shared 4 datasets are given in Table 9.

Regarding the same regression datasets used in `DTSemNet`, `DGT-Linear` and `TAO-Linear`, we also include these 5 shared datasets. We follow the same comparison setting and experimental setup as `DTSemNet` to include these 5 shared datasets for a more credible comparison against `DTSemNet`, `DGT-Linear` and `TAO-Linear`. Details of the shared 5 datasets are provided in Table 10.

For comparison of training optimality in Section 5.2, four small regression datasets with around 100 samples are also used. These datasets, collected from UCI and listed in Table 11, allow us to compare

the training optimality of our method against global optimal solutions. The global optimal method `ORT-MIP` is computationally solvable for such a small dataset size as observed in our experiments.

Besides, three two-dimensional synthetic regression datasets with 5000 samples each are also used to showcase the effectiveness of our tree optimization in approaching global training optimality. Synthetic datasets are advantageous as they hold a known ground truth. Based on tree-based decision rules with the form of `IF-THEN` statements, we design $2^D$ decision rules to create a dataset by the $D$-depth tree. These decision rules are used to assign samples to the corresponding leaf nodes along a sequence of branching tests. For instance, when generating a two-dimensional dataset for a 2-depth tree, one of the decision rules is described as `IF` $x_{i,1} + x_{i,2} > 0$ `and` $x_{i,1} - x_{i,2} > 0$, `THEN` $y_i = 0.5$. In this setting, we generate 3 synthetic datasets, termed "Syn-2", "Syn-3" and "Syn-4" . These datasets are created with varying tree depth settings of 2, 3, and 4, respectively.

Table 9: Group (ii) 4 regression datasets used in original work of soft trees [İrsoy et al., 2012].

| Dataset Index | Dataset Name | Dataset Size (n) | Feature Number (p) |
|---|---|---|---|
| SoftTree-1 | abalone | 4,177 | 8 |
| SoftTree-2 | puma8NH | 8,192 | 8 |
| SoftTree-3 | computer | 209 | 7 |
| SoftTree-4 | concrete | 103 | 7 |

Table 10: Group (ii) five shared regression datasets used in original work of `DTSemNet`, `DGT` and `TAO`.

| Dataset Index | Dataset Name | Dataset Size (n) | Feature Number (p) |
|---|---|---|---|
| DTSemNet-1 | Abalone | 4,177 | 8 |
| DTSemNet-2 | Comp-Active | 8,192 | 21 |
| DTSemNet-3 | Ailerons | 14,308 | 40 |
| DTSemNet-4 | CTSlice | 53,500 | 384 |
| DTSemNet-5 | YearPred | 515,345 | 90 |

Table 11: Group (iii) four small real-world regression datasets with about 100 samples.

| Dataset Index | Dataset Name | Dataset Size (n) | Feature Number (p) |
|---|---|---|---|
| Small-1 | concrete-slump-test-compressive | 103 | 7 |
| Small-2 | concrete-slump-test-flow | 103 | 7 |
| Small-3 | hybrid-price | 153 | 4 |
| Small-4 | lpga-2008 | 157 | 6 |

Regarding the 7 large-scale regression datasets each with at least 1,000,000 samples in Group **(iv)**, they are used to evaluate the scalability of our method. These datasets are listed in Table 12. The dataset size $n$ and the number of features $p$ are provided in the table.

Table 12: Group (iv) seven large-scale regression datasets each with at least 1 million samples.

| Dataset Index | Dataset Name | Dataset Size (n) | Feature Number (p) |
|---|---|---|---|
| Million-1 | BNG-Ailerons | 1,000,000 | 40 |
| Million-2 | BNG-cpu-act | 1,000,000 | 21 |
| Million-3 | BNG-cpu-small | 1,000,000 | 12 |
| Million-4 | BNG-puma32H | 1,000,000 | 32 |
| Million-5 | BNG-wisconsin | 1,000,000 | 32 |
| Million-6 | ACSPublicCoverage2018 | 1,138,289 | 19 |
| Million-7 | BNG-elevator | 1,000,000 | 18 |

## I The implementation Settings for Comparison Studies

To implement our "ReLU+Argmin"-based Differentiable Decision Tree optimization framework (`RADDT`), we utilize PyTorch that embeds auto differentiation tools and gradient-based optimizers. Our method is configured with $N_{epoch} = 3,000$ and $N_{start} = 10$, unless otherwise specified.

For benchmarking, the `Scikit-learn` library in `Python` is used to implement `CART` and random forest (`RF`) methods. `XGBoost` is implemented via their official package. The parameter values for these methods are set to default values, unless otherwise specified, such as the specific hyperparameters

tuning for `RF` and `XGBoost` discussed in Section 5.1 and Appendix J. The implementation of `HHCART`, `RandCART` and `OC1` is adapted from publicly sourced GitHub repository and programmed in `Python`. We modified their classification-oriented loss functions to adapt for regression tasks.

As for the local search method `ORT-LS`, we reproduce it in `Julia` due to the absence of open-source code for `ORT-LS`. Aiming to provide a reference for global training optimality, we also implement the MIP-based optimal oblique decision tree, `ORT-MIP`, in `Julia`, solving the MIP problem with `Gurobi` due to the absence of open-source code for `ORT-MIP`.

The `GradTree`, `SoftDT`, `DGT` and `LatenTree` methods are implemented using their respective open-source GitHub repositories, with adjustments made only to the epoch numbers to align with our methods. The `TEL` is implemented using their open-source code with their own default parameter settings, except for the number of trees and epoch numbers. We modify the epoch number to align with our experimental setup. Since it is a tree ensemble method, we set the number of trees to 10.

Since no open-source code is available for `TAO-Linear`, and the `DTSemNet` work directly uses the reported results from `TAO-linear` for comparison, we adopt the same approach to ensure consistency and credibility. `DTSemNet` itself is a decision tree with linear predictions. For consistency, it also adapts the original `DGT` tree with constant predictions into a linear variant, termed `DGT-Linear`. The datasets used in `TAO-Linear` are shared with `DTSemNet` and `DGT-Linear`, and `DTSemNet` directly collects the reported results from each method for comparison. Following their experimental setup, we apply the same data preprocessing, dataset splits and evaluation metric to implement our method on these five shared datasets. This allows for a fair and direct comparison between our results and the summarized benchmarks presented in `DTSemNet`' Table 4, covering `DGT-Linear`, `TAO-Linear`, and `DTSemNet` itself. Detailed implementation settings for these baselines can be found in `DTSemNet` Panda et al. [2024]. It is important to note that this, same-dataset comparison against `TAO-Linear`, `DGT-Linear` and `DTSemNet`, is the only experiment in which we directly adopt the reported results for `DGT` (`DGT-Linear`). In our other experiments involving `DGT`, including Table 3 in Section 5.2 and Table 22 in Appendix K.10, we implement the experiments using their open-source code software.

Experiments necessitating CPU computation were executed on the HPC Cluster, specifically utilizing "Dell EMC R440 CPU" configuration. Each CPU job is allocated 32G memory with a Time Limit of 7 days. Experiments for `ORT-MIP` requiring larger memory resources were carried out on the Oracle HPC Cluster, specifically with 2T memory and 128 cores. Concurrently, experiments requiring GPU resources were conducted on the "Narval" server, with an NVIDIA A100 GPU equipped.

# J  Hyperparameters Tuning for Tree Ensemble Methods

For a fair comparison, comprehensive hyperparameter tuning is also performed for three tree ensemble methods. Regarding random forest `RF`, the number of trees in a forest is a critical parameter. It is well-recognized that testing performance improves with an increase in the number of trees; however, the marginal gains become less pronounced as additional trees are added [Oshiro et al., 2012, Probst et al., 2019]. Accordingly, the number of trees is tuned across a set of $\{50, 100, 200, 300, 400, 500\}$. Moreover, the maximum tree depth is tuned over a broader range, from 1 to 50, to potentially capture optimal depth settings, given that `RF` empirically benefits from overly-deeper trees for enhanced testing accuracy. Other hyperparameters, including the number of features per split and the number of samples per tree, are maintained at default settings. These have been shown to balance the bias-variance trade-off, typically yielding robust performance with default values [Probst et al., 2019].

For `XGBoost`, we follow the same hyperparameter tuning schemes, varying the number of trees across a set $\{50, 100, 200, 300, 400, 500\}$, and the maximum tree depth from 1 to 50.

For the gradient-based tree ensemble `TEL`, no specific hyperparameter tuning is performed; we directly fix the number of trees to 10 and retain all other parameters at their default settings.

While our current hyperparameter tuning for tree ensembles, including those in Appendix K.6 with more-combination search for hyperparameters, is sufficient to support our main claims, especially given that our goal is not to claim superiority over ensembles, more tuning on regularization parameters could potentially yield even better results for tree ensembles.

# K The Complementary Results of Numerical Experiments

## K.1 Detailed testing accuracy comparison results on 17 medium-sized datasets in Group (i).

The testing accuracy ($R^2$) comparison for various decision trees on specific 17 medium-sized datasets in Group (i) is detailed in Table 13.

Table 13: Testing accuracy comparison for diverse decision trees on Group (i) 17 real-world datasets.

| Datasets | Greedy methods | | | | Gradient-based | | Heurisitc Local Search | Our Optimized Tree | |
|---|---|---|---|---|---|---|---|---|---|
| | CART | OC1 | RandCART | HHCART | GradTree | SoftDT | ORT-LS | RADDT | RADDT-Linear |
| 1 | 85.27 | 86.38 | 76.79 | 85.71 | 56.46 | 67.06 | 85.24 | 90.34 | 90.65 |
| 2 | 42.14 | 40.53 | 50.70 | 49.05 | 30.47 | 47.46 | 49.51 | 59.46 | 61.20 |
| 3 | 47.29 | 46.15 | 46.94 | 54.75 | 49.68 | 55.28 | 54.20 | 57.03 | 58.74 |
| 4 | 66.50 | 55.39 | 57.24 | 60.61 | 64.07 | 60.80 | 55.62 | 62.19 | 71.62 |
| 5 | 82.19 | 83.20 | 83.30 | 84.43 | 67.03 | 80.81 | 86.39 | 89.63 | 88.59 |
| 6 | 62.36 | 62.71 | 41.64 | 66.95 | 61.53 | 60.31 | 63.68 | 65.21 | 68.11 |
| 7 | 96.98 | 97.23 | 92.06 | 97.15 | 95.30 | 85.36 | 97.59 | 97.47 | 98.25 |
| 8 | 95.89 | 96.25 | 95.78 | 96.25 | 93.69 | 88.66 | 96.65 | 96.26 | 97.31 |
| 9 | 42.56 | 51.53 | 51.83 | 56.32 | 44.22 | 71.60 | 69.57 | 81.26 | 86.23 |
| 10 | 60.19 | 59.03 | 58.42 | 60.91 | 54.76 | 62.76 | 58.52 | 61.86 | 63.58 |
| 11 | 93.33 | 93.15 | 93.11 | 93.55 | 84.66 | 93.16 | 92.84 | 94.00 | 94.22 |
| 12 | 71.17 | 74.72 | 58.08 | 68.80 | 45.01 | 86.35 | 80.66 | 87.22 | 91.74 |
| 13 | 98.58 | 94.37 | 98.45 | 98.63 | 85.17 | 86.83 | 98.93 | 97.76 | 99.99 |
| 14 | 97.34 | 71.14 | 96.11 | 95.23 | 76.81 | 48.37 | 97.78 | 96.30 | 99.97 |
| 15 | 75.96 | 76.55 | 75.48 | 77.92 | 67.49 | 81.61 | 78.44 | 81.46 | 82.42 |
| 16 | 69.12 | 72.30 | 65.03 | 75.66 | 49.73 | 86.57 | 82.61 | 89.61 | 90.57 |
| 17 | 85.51 | 85.69 | 69.52 | 82.75 | 67.03 | 73.33 | 89.18 | 93.51 | 95.49 |

The testing accuracy ($R^2$) comparison for 3 tree ensembles on specific 17 medium-sized datasets in Group (i) is detailed in Table 14.

Table 14: Testing accuracy comparison for three tree ensembles on Group (i) 17 real-world datasets.

| Dataset Index | RF | XGBoost | TEL |
|---|---|---|---|
| 1 | 92.58 | 93.39 | 88.76 |
| 2 | 53.22 | 53.84 | 69.79 |
| 3 | 57.64 | 57.40 | 61.23 |
| 4 | 68.10 | 65.55 | 56.55 |
| 5 | 91.00 | 90.78 | 88.11 |
| 6 | 68.38 | 66.79 | 67.00 |
| 7 | 98.27 | 98.54 | 97.78 |
| 8 | 97.63 | 97.72 | 97.35 |
| 9 | 70.48 | 75.29 | 90.57 |
| 10 | 63.16 | 64.15 | 62.31 |
| 11 | 95.92 | 96.49 | 94.01 |
| 12 | 89.59 | 92.80 | 95.30 |
| 13 | 99.50 | 99.51 | 98.31 |
| 14 | 98.74 | 98.67 | 92.37 |
| 15 | 83.24 | 83.39 | 81.76 |
| 16 | 83.59 | 90.01 | 89.88 |
| 17 | 93.41 | 95.36 | 94.70 |

## K.2 Implementation details of same-dataset comparisons for certain baselines

For `GradTree`, we follow their experimental setup: using 20% of the data for testing, macro F1-score as evaluation metric, and averaging results over 10 runs with different random splits. Despite adhering to their dataset settings and hyperparameter configurations, we were unable to reproduce their reported results from Tables 1 and 2 in their paper. Therefore, we directly compare our `RADDT` results with their published results.

Regarding soft decision tree, no tabular datasets were used in [Frosst and Hinton, 2017], but we adopt the four tabular regression datasets from the original Soft Decision Tree study by [İrsoy et al., 2012]. As no specific dataset settings are provided in that work, we use our default experimental setup for both Soft Decision Tree and `RADDT`.

Regarding `TAO-Linear`, `DGT-Linear` and `DTSemNet`, since no open-source code is available for `TAO-Linear`, and the `DTSemNet` work directly uses the reported results from `TAO-Linear` for comparison, we adopt the same approach to ensure consistency and credibility. `DTSemNet` itself is a

decisoin tree with linear predictions. For consistency, it also adapts the original `DGT` tree with constant predictions to a linear variant, referred to as `DGT-Linear`. Following `DTSemNet`'s experimental setup, we apply the same data preprocessing, dataset splits (40% of the data for testing, unless otherwise specified for certain dataset) and evaluation metric (RMSE) to implement our method on the five shared datasets. This allows for a fair and direct comparison between our results and the summarized benchmarks presented in `DTSemNet`'Table 4, covering `DGT-Linear`, `TAO- Linear`, and `DTSemNet` itself. Detailed implementation settings for these baselines can be found in `DTSemNet`.

### K.3 Detailed results for same-dataset comparison on Group (ii) 27 datasets used by `GradTree`

Our same-dataset comparison for `GradTree` is performed on 27 of their original 36 datasets, with 9 excluded due to unavailability or ambiguity. Detailed dataset information is given in Appendix H. The result for each dataset is given in Table 15, using macro F1-score as the evaluation metric, following their original study. Our method outperforms `GradTree` by 4.99% on average in test performance.

Table 15: Testing comparison on 27 shared classification datasets in Group (ii) with `GradTree`

| Dataset Index | Dataset Name | Originally Reported Results in `GradTree` | Our `RADDT` |
|---|---|---|---|
| GradTree-1 | balance-scale | 59.30 | 90.10 |
| GradTree-2 | banknote-authentication | 98.70 | 99.70 |
| GradTree-3 | blood-transfusion | 62.80 | 66.00 |
| GradTree-4 | car-evaluation | 44.00 | 86.80 |
| GradTree-5 | congressional-voting-records | 95.00 | 96.60 |
| GradTree-6 | contraceptive-method-choice | 49.60 | 48.40 |
| GradTree-7 | dermatology | 93.00 | 94.40 |
| GradTree-8 | echocardiogram | 65.80 | 83.30 |
| GradTree-9 | iris | 93.80 | 95.10 |
| GradTree-10 | spambase | 90.30 | 92.10 |
| GradTree-11 | thyroid-disease-ann-thyroid | 90.50 | 95.80 |
| GradTree-12 | wine | 93.30 | 97.80 |
| GradTree-13 | adult | 74.30 | 76.40 |
| GradTree-14 | bank-marketing | 64.00 | 71.90 |
| GradTree-15 | credit-card | 67.40 | 68.20 |
| GradTree-16 | german | 59.20 | 66.10 |
| GradTree-17 | glass | 56.00 | 58.20 |
| GradTree-18 | heart-failure | 75.00 | 75.80 |
| GradTree-19 | landsat | 80.70 | 86.30 |
| GradTree-20 | loan-house | 71.40 | 73.20 |
| GradTree-21 | lymphography | 61.00 | 61.20 |
| GradTree-22 | mushrooms | 100.00 | 100.00 |
| GradTree-23 | raisins | 84.00 | 85.10 |
| GradTree-24 | segment | 94.10 | 88.80 |
| GradTree-25 | solar-flare | 15.10 | 15.40 |
| GradTree-26 | wisconsin-breast-cancer | 90.40 | 94.80 |
| GradTree-27 | zoo | 87.40 | 83.30 |

### K.4 Detailed results for same-dataset comparison on Group (ii) 4 datasets used by soft trees

We provide the comparison results for each dataset in Table 16. The results show that `RADDT` outperforms the soft decision tree by 6.88% in testing accuracy ($R^2$), again aligning with prior results.

Table 16: Testing comparison on 4 shared regression datasets in Group (ii) with Soft Decision Tree

| Dataset Index | Dataset Name | Soft Decision Tree | Our `RADDT` |
|---|---|---|---|
| SoftTree-1 | abalone | 55.28 | 57.03 |
| SoftTree-2 | puma8NH | 60.31 | 65.21 |
| SoftTree-3 | computer | 57.01 | 56.69 |
| SoftTree-4 | concrete | 25.93 | 47.10 |

### K.5 Detailed results for same-dataset comparison on Group (ii) 5 datasets used by `DGT-Linear`, `TAO-Linear` and `DTSemNet`

As earlier discussed in Section 5.1, we have compared our method with `DGT-Linear`, `TAO-Linear` and `DTSemNet` on the five shared datasets as used in the original work of `DTSemNet`. We extend `DTSemNet`'s Table 4 by including our results under the same experimental setup in Table 17.

Table 17: Comparison on Group (ii) five datasets used by `TAO-Linear`, `DGT-Linear` and `DTSemNet`

| Datasets | Our RADDT-Linear [1] | | DTSemNet [2] | | DGT-Linear [2] | | TAO-Linear [2] | |
|---|---|---|---|---|---|---|---|---|
| | RMSE | Rank | RMSE | Rank | RMSE | Rank | RMSE | Rank |
| Abalone | **1.984** | 1 | 2.135 | 3 | 2.144 | 4 | 2.07 | 2 |
| Comp-Active | **2.519** | 1 | 2.645 | 3 | 2.645 | 3 | 2.58 | 2 |
| Ailerons | **1.154** | 1 | 1.66 | 2 | 1.67 | 3 | 1.74 | 4 |
| CTSlice | **1.138** | 1 | 1.45 | 3 | 1.78 | 4 | 1.16 | 2 |
| YearPred | 11.934 | 4 | **8.99** | 1 | 9.02 | 2 | 9.08 | 3 |

[1] Following `DTSemNet` work, trees with linear prediction are compared. [2] These results are taken directly from `DTSemNet`'s Table 4.

## K.6 More extensive hyperparameter tuning for tree ensembles on Group (i) datasets

As previously indicated in Table 2, the findings of our methods being comparable to tree ensembles `RF` and `XGBoost` are based on 300-combination specific hyperparameter tuning scheme. In that setting, our method is tuned only over tree depths from 1 to 12 with 12 combinations, ensuring a much lighter tuning for fairness.

To further support this finding, we extend the parameter search space to 5,400 combinations. This involves tuning the "tree depth" within a range of {1-50, 55, 60, 70, 80, 90, 100, 150, 200, 250, 300}, "the number of trees" within {5, 10, 50, 100, 200, 300, 400, 500, 600, 700}, and with the additional tuning for `RF` on "the minimum samples per leaf and per split" in {1, 2, 4}, and for `XGBoost` on "subsample" and "colsample_bytree" in {0.6, 0.8, 1.0}. Additionally, to provide even stronger evidence, we further use PyCaret automated machine learning tool to perform an extensive 10,000-combination hyperparameter tuning for `XGBoost`, exploring its whole hyperparameter space with PyCaret's default search settings.

As shown in Table 18, the 5,400-combination tuning yields slight improvements in testing accuracy for both `RF` and `XGBoost`, with 0.11% gain for `RF` and 0.26% gain for `XGBoost` over the 300-combination tuning in Table 2. The extensive 10,000-combination tuning for `XGBoost` via PyCaret leads to just a 1.04% gain. Despite extensive tuning, our key finding remains consistent: our single tree achieves accuracy comparable to ensemble methods. Notably, our aim is not to claim superiority over tree ensembles, as any base learner including decision tree can be improved via ensemble techniques. Comparing a single tree to tree ensembles is inherently unfair. Instead, our goal is to showcase the potential of a single, optimized tree like ours in achieving high testing accuracy.

Table 18: Accuracy comparison on Group (i) datasets with extensive tuning for tree ensembles.

| The Number of Hyperparameter Tuning Combinations | Testing Accuracy ($R^2$, %) |
|---|---|
| RF (300-combination reported in Table 2) | 82.62 |
| RF (5400-combination) | 82.73 |
| XGBoost (300-combination reported in Table 2) | 83.51 |
| XGBoost (5400-combination) | 83.77 |
| XGBoost (10000-combination by PyCaret) | 84.55 |

## K.7 Detailed comparison against optimal trees on 4 small datasets in Group (iii)

As earlier discussed, `ORT-MIP` [Bertsimas and Dunn, 2017] is computationally infeasible for previously-used datasets, even with 128 cores and a two-day time limit. It can only solve the 2-depth tree training problem on 4 small datasets in Group (iii) to a global optimum with 1% gap. Due to these limitations, only 50% samples are used for training in this experiment. Detailed training accuracy comparisons against `ORT-MIP` on these 4 small datasets are presented in Table 19.

Table 19: Train accuracy comparison results on solvable 4 small datasets in Group (iii).

| Dataset | Sample | Feature | ORT-MIP | RADDT |
|---|---|---|---|---|
| concrete-slump-test-compressive | 103 | 7 | 87.79 | 87.12 |
| concrete-slump-test-flow | 103 | 7 | 89.89 | 84.92 |
| hybrid-price | 153 | 4 | 78.09 | 73.50 |
| lpga-2008 | 157 | 6 | 86.79 | 85.73 |
| Average Training Accuracy ($R^2$, %) | | | 85.64 | 82.82 |

Regarding the optimal sparse trees [Zhang et al., 2023], it complements earlier work on optimal trees (`ORT-MIP`) that fails to consider sparsity. As for experimental comparisons, optimal sparse trees

may not be directly comparable to our method. First, optimal sparse trees are limited to axis-aligned splits, which generally yield lower accuracy than oblique trees. Second, they are designed for binary features, requiring feature discretization that may degrade performance on our continuous-feature datasets. We therefore choose not to include this in our experiments, as it could unfairly disadvantage their method under our evaluation setting.

### K.8 Detailed comparisons against known ground truth on 3 synthetic datasets in Group (iii)

We provide the detailed comparison results for each synthetic dataset in Table 20. The known ground truths for both train and test accuracy are 100%. The results show that our `RADDT` closely approximate global optimality in train and test accuracy, with 0.64% difference over ground truth on average. These results, together with previous global solution comparisons, further validate the effectiveness of our tree optimization.

Table 20: Training and Testing performance on synthetic datasets in Group (iii).

| Dataset | Fitted Depth | | Training Accuracy ($R^2$, %) | | Testing Accuracy ($R^2$, %) | |
|---|---|---|---|---|---|---|
| | Ground-truth | RADDT | Ground-truth | RADDT | Ground-truth | RADDT |
| Syn-2 | 2 | 2 | 100 | 99.96 | 100 | 99.99 |
| Syn-3 | 3 | 3 | 100 | 98.78 | 100 | 98.71 |
| Syn-4 | 4 | 4 | 100 | 99.35 | 100 | 99.39 |

### K.9 Ablation experiments on the strategies used in our tree optimization

The observed superiority in training accuracy of our `RADDT` method can be attributed to two key strategies designed to enhance approximation accuracy, as discussed in Section 4: the multi-run warm start annealing strategy, and the strategy of adjusting initial solution for gradient-based optimization. The comparative results, presented in Table 21, clearly illustrate the impact of these strategies on `RADDT`. It should be noted that our method, which utilizes the multi-run warm start annealing, is referred to as `RADDT` in the table. In contrast, when this strategy is not employed, the method is denoted by specific values of scale factors, such as $\alpha = 1$ and $\alpha = 150$. Additionally, the strategy of adjusting initial solutions is evaluated in the table, with the method labeled as `RADDT` without adjustments to initial solution and `RADDT` with adjustments to initial solution.

Regarding multi-run warm start annealing, we analyze training accuracy with and without this strategy. Without this strategy, we utilize fixed scale factors for comparison: a standard softmin function with a small $\alpha = 1$, and a larger scale factor $\alpha = 150$. The findings indicate substantial improvements with the annealing approach: training accuracy increased by 9.7%, 9.32%, 8.46% and 1.07% at tree depths of 2, 4, 8, and 12, respectively, when compared to the standard softmin function without any scaling. Moreover, when compared to fixed scale factors larger than 1, the annealing strategy also yields a higher training accuracy. These results confirm that our multi-run warm start annealing strategy is effective in enhancing training accuracy, by striking a greater balance between the approximation accuracy of scaled softmin function and the stability of gradient-based optimization.

The strategy of adjusting the initial solution for gradient-based optimization also contributes to these improvements. It boosts training accuracy by 1.3%, 2.89%, 1.01% and 1.84% at tree depths of 2, 4, 8, and 12, respectively. Notably, these gains are achieved solely through improved initialization, without involving in the optimization process.

Table 21: The effectiveness of strategies used in our tree optimization approach on training accuracy.

| Item | | $D = 2$ | $D = 4$ | $D = 8$ | $D = 12$ |
|---|---|---|---|---|---|
| without Multi-run Warm Start Annealing | $\alpha = 1$ (Standard Softmin Function) | 60.78 | 69.97 | 81.46 | 93.45 |
| (A fixed Scale Factor) | $\alpha = 150$ | 68.00 | 78.31 | 88.08 | 94.42 |
| with Multi-run Warm Start Annealing | RADDT without adjustments to initial solution | 70.48 | 79.29 | 89.92 | 94.52 |
| (Our RADDT method) | RADDT with adjustments to initial solution | 71.78 | 82.18 | 90.93 | 96.36 |

### K.10 Scalability comparison on 7 million-scale datasets in Group (iv)

In Section 5.3, we analyze the scalability of various methods on 7 million-scale datasets in Group (iv) under a fixed-depth setting. The detailed results are presented in Table 22.

Table 22: Scalability comparison under fixed-depth setting on 7 million-scale datasets in Group (iv).

| Methods[1] | Training Accuracy ($R^2$, %) | | | | Testing Accuracy ($R^2$, %) | | | | Training Time (s) | | | |
|---|---|---|---|---|---|---|---|---|---|---|---|---|
| | D2 | D4 | D8 | D12 | D2 | D4 | D8 | D12 | D2 | D4 | D8 | D12 |
| CART | 24.18 | 35.19 | 42.06 | 47.80 | 23.91 | 34.99 | 41.35 | 41.57 | 6 | 9 | 17 | 25 |
| RandCART | 17.17 | 22.71 | 28.35 | 33.22 | 17.11 | 22.57 | 27.98 | 29.78 | 9 | 16 | 38 | 67 |
| HHCART | 22.40 | 27.20 | 36.40 | 42.29 | 22.29 | 27.07 | 36.03 | 38.79 | 12 | 24 | 46 | 84 |
| GradTree | 19.67 | 26.92 | 33.36 | / | 19.48 | 26.60 | 33.16 | / | 1,387 | 3,058 | 14,570 | / |
| SoftDT | 18.19 | 27.36 | 28.36 | / | 18.13 | 27.24 | 28.21 | / | 29 | 136 | 16,263 | / |
| DGT | 30.26 | 35.27 | / | / | 30.10 | 34.99 | / | / | 202,537 | 232,574 | / | / |
| LatentTree | 33.35 | 40.04 | / | / | 33.00 | 39.74 | / | / | 73,965 | 72,130 | / | / |
| RADDT | 33.93 | 39.39 | 43.56 | 51.67 | 33.56 | 39.01 | 42.27 | 39.59 | 8,136 | 5,596 | 21,874 | 80,731 |

[1] OC1 and ORT-LS fail even at depth 2 within 14 days. GradTree, SoftDT, DGT, and LatentTree face out-of-memory issues when scaling, with GradTree and SoftDT failing at depth 12, and DGT and LatentTree only solvable at depths 2 and 4.

### K.11 GPU-accelerated implementation for our RADDT experiments

Our differentiable tree is trained by solving an unconstrained optimization problem, offering substantial scalability benefits via its reformulated structure. Furthermore, by leveraging mature frameworks like PyTorch, our implementation facilitates GPU acceleration, including distributed data parallel with multi-GPU. Under ideal full GPU utilization, training can also be distributed across multiple GPUs, further reducing time in proportion to the number of GPUs used. Concerning the training time reported in Table 3, our RADDT method achieves superior accuracy without a substantial increase in training time. While single-GPU acceleration is enabled for both our RADDT and other compared gradient-based trees, Our RADDT attains more efficient acceleration by fully leveraging matrix operation in computing our designed loss in Equation (7), which integrates unique tree path formulation. Specifically, by introducing the parameter $U_{i,t}$, we store the deterministic sample route and assignments in a matrix to avoid loop-based calculation at every iteration, enabling efficient matrix operations for loss and gradient calculations. Additionally, for training 12-depth trees on Group (i) datasets with more than 10,000 samples, we use 4 GPUs to further accelerate our RADDT.

Regarding scalability experiments on million-scale datasets reported in Table 22, we utilize 4 GPUs to train our RADDT at depths 2, 4, and 8, and 8 GPUs for depth 12. This distributed data parallelism is easily implemented using our extensible framework, which supports multi-GPU training through simple commands without requiring any specialized parallelization design.

### K.12 Training time and speedup analysis on Group (i) datasets with over 10,000 samples

As in Table 3, our RADDT achieves superior accuracy without substantial increase in training time compared to other trees, such as the second-best ORT-LS. These scalability advantages become more pronounced when dealing with larger matrix operations, particularly involving in deeper trees or larger datasets with more samples and feature dimensions. To illustrate this, we provide a detailed comparison of training time and speedup for 12-depth tree training between RADDT and ORT-LS on the datasets with more than 10,000 samples in Group (i). The results are summarized in Table 23.

Table 23: The speedup of RADDT over ORT-LS for 12-depth tree training on Group (i) datasets with more than 10,000 samples.

| Dataset Index | Dataset Name | Dataset Size (n) | Feature Number (p) | ORT-LS (Time, s) | RADDT (Time, s) | Speedup (ORT-LS/RADDT) |
|---|---|---|---|---|---|---|
| 12 | electrical-grid-stability | 10,000 | 12 | 197,361.31 | 1,691.10 | 116.71 |
| 13 | condition-based-maintenance_compressor | 11,934 | 16 | 232,929.77 | 2,036.77 | 114.36 |
| 14 | condition-based-maintenance_turbine | 11,934 | 16 | 251,017.34 | 2,026.15 | 123.89 |
| 15 | ailerons | 13,750 | 40 | 965,968.22 | 2,236.61 | 431.89 |
| 16 | elevators | 16,599 | 18 | 485,509.64 | 2,622.17 | 185.16 |
| 17 | friedman-artificial | 40,768 | 10 | 671,460.77 | 4,331.58 | 155.02 |

### K.13 Comparison of our method's training time in GPU versus CPU settings

In this section, we provide a comparison of our method's training time in GPU versus CPU settings, although the advantages of GPU acceleration are expected. We use a fixed dataset, "sgemm-product", with subsets varying sizes from {100, 1,000, 5,000, 10,000, 50,000} for training 8-depth and 12-depth trees. As expected, GPU training is significantly faster. For instance, on 50,000 samples with an 8-depth tree, GPU training is 117 times faster than CPU. For a 12-depth tree on 1,000 samples, we

observe a 107 times speed-up. On larger datasets and deeper trees, the CPU version often fails to complete within 5 days. This highlights the practical importance of GPU acceleration for scalability.

Table 24: Training time comparison of our `RADDT` method in GPU versus CPU settings.

| | Training Time (Second) | | | |
|---|---|---|---|---|
| | 8-depth treee | | 12-depth tree | |
| Datasize | One GPU | CPU | One GPU | CPU |
| 100 | 337 | 615 | 769 | 10,912 |
| 1,000 | 359 | 2,938 | 1,043 | 111,111 |
| 5,000 | 466 | 22,082 | 2,799 | OOT >5 days |
| 10,000 | 614 | 46,132 | 5,057 | OOT >5 days |
| 50,000 | 2,949 | 345,992 | 41,564 | OOT >5 days |

Additionally, as discussed in Appendix K.10, Appendix K.11 and Appendix K.12, also observed in this comparison, the GPU advantages become increasingly pronounced with larger matrix operations, particularly involving deeper trees, larger datasets with more samples and feature dimensions.

### K.14 Interpretability analysis of our optimized tree

We fully agree that orthogonal trees like CART are more interpretable at the same, fixed depth. However, as originally discussed in `OC1` [Murthy et al., 1994], oblique trees can become more interpretable when they require substantially less depth.

Our `RADDT`'s interpretability can be assessed from three aspects: tree-based prediction logic, unique prediction path and the complexity of decision rules. First, `RF`, in Table 2, using an average of 314.71 trees for higher accuracy, almost losing the interpretability for final predictions compared to a single tree. Second, our tree with ReLU-based hard splits maintains a unique decision path leading to final predictions. These hard splits preserve the interpretability of True-False decision-making, which is often compromised in soft trees with a probabilistic path. Third, regarding the complexity of decision rules, oblique trees tend to generate smaller trees with clear oblique decision boundaries compared to orthogonal trees like `CART`. For example, using synthetic dataset "Syn-4" in Table 20, `CART` achieves only 85.27% training accuracy for a 4-depth tree, whereas our `RADDT` attains 99.35% for the same depth. To achieve a comparable train accuracy, `CART` requires a deeper 9-depth tree with accuracy reaching to 99.88%. Detailed results of `CART` under various depths are given in Table 25. `CART` with 9-depth tree results in a complex, staircased boundary, shown in Figure 7. Such complexity can make decision rules difficult to interpret, whereas oblique trees with simpler and fewer splitting rules. It makes our `RADDT` more interpretable despite the use of 2-feature combinations in tree splits. Notably, this example does not imply oblique tree holds better interpretability, but rather to exhibit exclusive interpretability in certain datasets where underlying data distribution align with oblique boundaries.

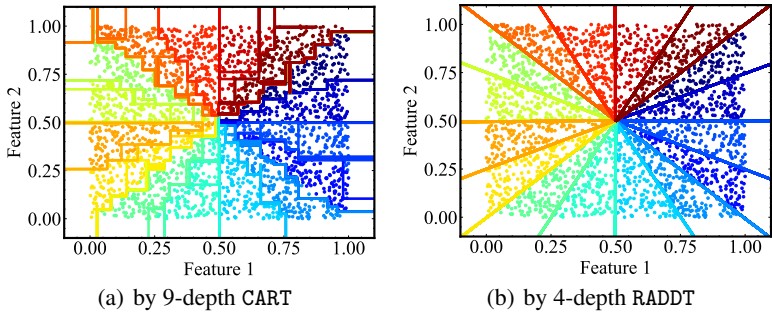

(a) by 9-depth `CART`          (b) by 4-depth `RADDT`

Figure 7: The decision boundary comparison for dataset "Syn-4".

Table 25: Train and Test accuracy for `CART` on synthetic dataset "Syn-4" under various tree depths.

| Item | Various Tree Depth | | | | | | | | | |
|---|---|---|---|---|---|---|---|---|---|---|
| | 1 | 2 | 3 | 4 | 5 | 6 | 7 | 8 | 9 | 10 |
| Training Accuracy ($R^2$, %) | 27.65 | 56.86 | 73.39 | 85.27 | 92.41 | 96.49 | 98.20 | 99.61 | 99.88 | 99.96 |
| Testing Accuracy ($R^2$, %) | 25.91 | 54.23 | 71.67 | 80.65 | 84.36 | 87.04 | 87.93 | 87.83 | 87.75 | 88.19 |

When compared to other oblique trees as reported in Table 1 and Table 2, our methods achieve superior accuracy but also with a lower depth, such as our `RADDT` with depth of 7.29 and `RADDT-Linear` with depth of 4.82. This reduction in depth significantly enhances interpretability with fewer rules. To illustrate, a 2-depth tree yields 4 decision rules across 2 layers, whereas a 10-depth tree produces 1024 rules across 10 layers. Understanding hundreds of nested IF-THEN rules can be challenging.

Additionally, as in Table 4, our methods can achieve slightly better accuracy than `CART` with significantly smaller trees: average depths of 2.82 (`RADDT`) and 1 (`RADDT-Linear`), compared to 10.24 for `CART`, leading to 30-80 times fewer parameters. In this case, for example, 1-depth tree consists of only 2 leaf nodes with 2 rules, which is highly interpretable because we can explain model predictions as "IF the feature combination is less than a threshold, THEN the prediction is based on the left leaf node; otherwise, right leaf node." These results highlight oblique tree's advantages in higher accuracy with smaller depth, typically yielding less prediction rules and fewer model parameters.

Therefore, interpretability depends on multiple factors including tree depth, feature number, and the accuracy-simplicity trade-off. There is no interpretability superiority of one approach over the other, as it depends on specific application scenarios.

### K.15 Overfitting issue and comparison for trees with linear predictions

As discussed in Section 5.6, overfitting issues have been observed with both our `RADDT` and `RADDT-Linear` method. Upon further comparison of our `RADDT-Linear` with the existing open-source library `linear-tree`, it is evident that the overfitting issues are more pronounced in trees with linear predictions. The open-source software `linear-tree` exhibits significantly more severe overfitting issues at depths such as 8 and 12, as detailed in Table 26.

For our `RADDT-Linear`, although the training accuracy improves by 6.52% from depth 4 to 8, testing accuracy conversely drops by 9.67%, indicating a serious overfitting issue. To mitigate the overfitting issues for `RADDT-Linear`, we preliminarily attempt to apply $L_1$ regularization to trainable variables $\boldsymbol{A}$. This approach involves incorporating a regularization term into the loss function $\mathcal{L}$ below, serving to penalize the complexity of the tree structure. The regularized loss is delineated in Equation (8), where $\lambda$ denotes the regularization strength and $\| \cdot \|_1$ represents the $L_1$ norm.

However, identifying the appropriate regularization strength $\lambda$ proves to be challenging during our experiments, necessitating extensive hyperparameter tuning. This tuning significantly increases the computational cost and the implementation complexity of our method. Consequently, in our experiment reported in Table 2, we did not tune this parameter. We just use a very small value $1e-4$ to implement a minimal regularization, aiming to enhance testing accuracy without greatly compromising the optimization capabilities. With this slight regularization, the testing accuracy of `RADDT-Linear` reported in Table 2 is improved by 0.58% compared to the results without regularization, as shown in Table 27. Despite the slight improvement, the overfitting issues for `RADDT-Linear` still exist, and our preliminary regularization is not sufficient to address this issue. Significant improvements in testing accuracy are achievable through appropriate regularization strategies; however, this requires further exploration and is limited in this paper.

Table 26: Comparison of training and testing accuracy for `RADDT-Linear` and `linear-tree`.

| Depth | Training Accuracy ($R^2$, %) | | Testing Accuracy ($R^2$, %) | |
|---|---|---|---|---|
| | linear-tree | RADDT-Linear | linear-tree | RADDT-Linear |
| 2 | 79.85 | 82.92 | 79.46 | 81.74 |
| 4 | 85.47 | 86.73 | 73.58 | 83.10 |
| 8 | 93.33 | 93.25 | -376648.84 (overfitting) | 73.43 |
| 12 | 98.31 | 98.00 | -11288345.94 (overfitting) | 38.98 |

$$\mathcal{L}_{reg} = \sum_{i=1}^{n} \sum_{t \in \mathbb{T}_L} \mathbb{S}\left(U_{i,t}\right) \left(y_i - \left(\boldsymbol{k}_t^T \boldsymbol{x}_i + h_t\right)\right)^2 + \lambda \sum_{t \in \mathbb{T}_b} \|\boldsymbol{a}_t\|_1 \tag{8}$$

Table 27: The comparison for `RADDT-Linear` with and without regularization across 17 datasets.

| Item | RADDT-Linear without regularization | RADDT-Linear with regularization |
|---|---|---|
| Testing Accuracy ($R^2$, %) | 84.05 | 84.63 |

