# OpenReview forum: "Differentiable Decision Tree via "ReLU+Argmin" Reformulation"
_NeurIPS.cc/2025/Conference — NeurIPS 2025 spotlight_

### Official Review · Reviewer_Mko3 · 2025-06-08

**Clarity:** 3
**Significance:** 3
**Originality:** 3
**Rating:** 5
**Confidence:** 3

**Summary:**

The authors introduce a method to fit differentiable, oblique decision trees using ReLU+Argmin optimization. The method relies on a warm-start annealing scheme and a softmin approximation during optimization. The procedure can be sped up using GPU parallelization. Experiments against 14 tree-based baselines show performance improvements and speed-ups relative to optimal oblique trees for fairly large trees (up to depth 12) and fairly large datasets (e.g. 1 million samples). Overall the approach and results seem sound.

**Questions:**

- How does the inference time for the authors' proposed approach compare to the inference time for baseline models such as CART?
- As the authors' method appears under-regularized, the authors may want to try [hierarchical shrinkage](https://arxiv.org/abs/2202.00858) on their trees

**Ethical Concerns:**

["NO or VERY MINOR ethics concerns only"]

**Final Justification:**

I think the authors for their clarifications and maintain my accept score of 5.

**Limitations:**

The interpretability of the proposed approach is not fully clear and the authors' discussion of it (Sec 5.4) does not seem to admit this. While RADDT can provide some interpretability, there is certainly a gap between RADDT and a concise axis-aligned decision tree (as suggested by Appendix A).

**Quality:**

3

**Strengths And Weaknesses:**

- S1: The author's problem is well motivated and the paper is well-written
- S2: The authors' experiments are comprehensive and show strong improvements
- S3: the authors' code appears fairly straightforward to use
- W1: Much of the authors' speed-up comes from using relatively expensive GPU computation. Can the authors show how the method compares when using CPUs? Also, It would also be nice to see a quantitative comparison against baselines that are also amenable to GPU speed-up (e.g. a soft tree method, e.g. Hzeimeh et al. 2020 or Frosst & Hinton 2017).
- W2: The authors are missing some discussion of related work, e.g. optimal axis-aligned trees such as [GOSDT](https://arxiv.org/abs/2006.08690).

---

> ### Author Rebuttal · Authors · 2025-07-31
>
> We sincerely thank the reviewer for recognizing the contributions of our work and for providing constructive feedback and insightful questions. Below, we address each of the comments in detail.
>
> >**Q1-"Much of the authors' speed-up comes from using expensive GPU computation. Can the authors show how the method compares when using CPUs? Also, It would also be nice to see a quantitative comparison against baselines that are also amenable to GPU speed-up (e.g. a soft tree Hazimeh et al. 2020 or Frosst & Hinton 2017)."**
>
> Thank you for your thoughtful comment on GPU speedup. We would like to provide additional experiments and clarifications below.
>
> To address the first part of your question, we conduct additional training time experiments for our method on both CPU and GPU. We use a fixed dataset with subsets varying sizes from {100, 1k, 5k, 10k, 50k} for training 8-depth and 12-depth trees. As expected, GPU training is significantly faster. For instance, on 50k samples with an 8-depth tree, GPU training is 117 times faster than CPU. For a 12-depth tree on 1k samples, we observe a 107× speed-up. On larger datasets and deeper trees, the CPU version often fails to complete within 5 days. This highlights the practical importance of GPU acceleration for scalability.
>
> Additionally, as discussed in appendix K.8 and K.9 and also observed in this new comparison results, the advantages of GPU acceleration become increasingly pronounced with larger matrix operations, particularly involving deeper trees, larger datasets with more samples and feature dimensions.
>
> |  |8-depth tree|  |12-depth tree|  |
> |-|-|-|-|-|
> |Datasize|GPU (second)|CPU (second)|GPU (second)|CPU (second)|
> |100|337|615|769|10,912|
> |1,000|359|2,938|1,043|111,111|
> |5,000|466|22,082|2,799|OOT>5 days|
> |10,000|614|46,132|5,057|OOT>5 days|
> |50,000|2,949|345,992|41,564|OOT>5 days|
>
> Regarding your next question on GPU comparison with SoftDT (Frosst & Hinton, 2017) and TEL (Hazimeh et al., 2020), we would like to clarify that SoftDT already supports GPU acceleration in its original code.
> We directly use their GitHub code without modification to ensure fairness. Their code utilizes GPU acceleration. As shown in Table 4, at smaller tree depths like 4, SoftDT is faster than our method. However, at larger depths like depth 12, our method achieves a 21× speedup compared to SoftDT.
>
> For TEL, we also use their original code, which does not provide GPU acceleration but leverages C++ kernels for efficient forward and backward propagation.
>
> We hope this clarifies the comparison and addresses your concerns.
>
> >**Q2-"The authors are missing some discussion of related work, e.g. optimal axis-aligned trees such as GOSDT."**
>
> Thank you for bringing this excellent work to our attention. GOSDT represents a key advancement in optimal sparse decision trees and complements earlier work on optimal axis-aligned trees without considering sparsity. We will cite it in the revised introduction to provide a more complete overview of optimal decision trees.
>
> Regarding experimental comparison, GOSDT may not be directly comparable to our method. First, it is limited to axis-aligned splits, which generally yield lower accuracy than oblique trees. Second, GOSDT is designed for binary features, requiring feature discretization that may degrade performance on our continuous-feature datasets. We therefore choose not to include GOSDT in our experiments, as it could unfairly disadvantage their method under our evaluation setting. We will clarify this rationale in the revised manuscript.
>
> >**Q3-"How does the inference time for the authors' proposed approach compare to the inference time for baseline models such as CART?**
>
> Thank you for your insightful comment on inference time, which greatly helps assess model complexity. **We compare the inference time of our method with CART from two perspectives**: **(a)** models with the reported accuracy in the manuscript (GET at 82.39%, GET-Linear at 84.63%, and CART at 74.85%); **(b)** models with similar accuracy (GET at 75.23%, GET-Linear at 77.21%, and CART at 74.85%). The second perspective is particularly relevant, as our models can match or exceed CART's accuracy while using much smaller trees. This enables us to explore more model variants by trading off a reduction in accuracy for significantly reduced inference time and model size. To ensure fairness, we average the inference time for each dataset over 10,000 repetitions and report the mean inference time across all datasets in the table below.
>
> - (a) As analyzed in our response to Reviewer oByt for Q1, CART achieves 74.85% accuracy with an average depth of 10.24 and approximately 3,141 parameters. In comparison, GET reaches 82.39% accuracy with a depth of 7.29 and ~10,504 parameters, while GET-Linear reaches 84.63% accuracy at depth 4.82 with ~4,045 parameters. In this setting, our methods involve more parameters than CART, and thus slightly higher inference time.
>
> - (b) However, if we aim to achieve accuracy comparable to CART (74.85%), our new comparison results show that the required depth of our trees can be significantly reduced, to 2.82 for GET and 1.0 for GET-Linear, resulting in only 101.71 and 40.47 total parameters on average, respectively. In this setting, GET still improves upon CART by 0.38%, while GET-Linear exceeds it by 2.36%. This yields a 31-fold and 78-fold parameter reduction compared to CART, respectively, with both models maintaining superior accuracy. **As a result, the inference time of GET becomes comparable to CART, while GET-Linear is even faster.** Note that all implementations use Python and PyTorch, where inference time may not scale strictly with parameter count due to framework overhead. In a pure C++ implementation, it would likely align more closely with parameter count.
>
> |Models with different accuracy|Tree Depth|Total Number of Parameters|Inference Time (second)|
> |-|-|-|-|
> |CART (74.85)|10.24|3,141|0.00044|
> |Our GET (82.39)|7.29|10,504|0.00188|
> |Our GET-Linear (84.63)|4.82|4,045|0.00161|
> |Our GET (75.23)|2.82|102|0.00054|
> |Our GET-Linear (77.21)|1|40|0.00039|
>
> In summary, our methods does not necessarily require more parameters for improved accuracy compared to axis-aligned trees like CART. The total inference cost depends on the desired trade-off between accuracy and model size. Therefore, we utilized these two distinct perspectives (a) and (b) to offer a comprehensive comparison of the inference time between our method with CART. We will incorporate this clarification into the revised manuscript.
>
> >**Q4-"As the authors' method appears under-regularized, the authors may want to try hierarchical shrinkage on their trees"**
>
> Thank you for suggesting this important study. Hierarchical shrinkage is an efficient post-hoc regularization technique for trees. We will cite this work in our revised manuscript as part of the regularization discussion and consider its integration in future research.
>
> >**Q5-"The interpretability of the proposed approach is not fully clear and the authors' discussion of it (Sec 5.4) does not seem to admit this. While RADDT can provide some interpretability, there is certainly a gap between RADDT and a concise axis-aligned decision tree"**
>
> Thank you for your insightful comment. We are sorry for any confusion caused by the current discussion and will revise the manuscript to better clarify interpretability through both illustrative examples and empirical evidence.
>
> We agree that axis-aligned trees like CART are more interpretable at the same depth. However, as originally discussed in OC1 (Murthy et al., 1994), oblique trees can become more interpretable when they require substantially less depth. For instance, on a synthetic 2-feature dataset (Appendix K.10, Fig. 7), our method produces a depth-4 oblique tree with 16 IF–THEN rules, whereas CART needs depth-9 and 512 rules, making our tree more interpretable despite the use of 2-feature combinations.
>
> Additionally, as noted in the response to your Q3, our models can also achieve slightly better accuracy than CART with significantly smaller trees: average depths of 2.82 (GET) and 1 (GET-Linear), compared to 10.24 for CART, leading to 30–70 times fewer parameters. In this case, for example, 1-depth tree consists of only 2 leaf nodes with 2 rules, which is highly interpretable because we can explain the model predictions as "IF the feature combination is less than a threshold, THEN the prediction is based on the left leaf nodes; otherwise, right leaf nodes." These experimental results highlight oblique tree's advantages in higher accuracy with smaller depth, typically yielding less prediction rules and fewer model parameters.
>
> Therefore, interpretability is not strictly tied to axis-aligned splits. It depends on multiple factors including tree depth, feature number, and the accuracy–simplicity trade-off, which may vary across different application scenarios. There is no interpretability superiority of one approach over the other, as it depends on the application context.

---

> > ### Comment · Reviewer_Mko3 · 2025-08-01
> >
> > I think the authors for their clarifications and maintain my accept score of 5.

---

> > > ### Author Response · Authors · 2025-08-02
> > >
> > > We sincerely thank the reviewer for their time and thoughtful evaluation. We are glad to hear that our clarifications helped address your concerns. We greatly appreciate your recognition of our work and the insightful feedback provided throughout the review process, which has contributed meaningfully to improving the quality of our paper.
> > >
> > > Once again, we sincerely thank you for your recognition, encouragement, and thoughtful support.

---

### Official Review · Reviewer_oByt · 2025-06-10

**Clarity:** 4
**Significance:** 3
**Originality:** 3
**Rating:** 5
**Confidence:** 3

**Summary:**

This paper introduces RADDT, a binary differentiable tree which uses oblique decisions. This model leverages differentiability in two steps: (i) it describes paths by counting the number of rule violations through relu functions (where a null output means a correct path) and (ii) it uses a softmin function to mimick the argmin operator for identifying the unique path leading to a decision. The model shows competitive performances to 14 (non-)differentiable trees with regards to both classification and regression tasks on an extensive benchmark of 67 datasets.

**Questions:**

Following the points highlighted in weaknesses, here are the isolated questions:

1. Could the authors mitigate the cost of their accuracy improvements depending on the number of parameters compared to other models?
2. Could the authors provide better insight on the training time specifically depending on the number of samples, features and tree depth?
3. Could the authors further detail the nature of the initialisation SVM and how frequently and harmful degenerate cases happen?
4. What are the authors' stance on the interpretability of their (non-sparse) oblique tree when depth, a user-defined hyperparameter, exceeds 5?

**Ethical Concerns:**

["NO or VERY MINOR ethics concerns only"]

**Final Justification:**

I have decided to give a final score of 5 to this paper. The authors addressed my questions and concerns within the limit of what was reasonable and in scope with respect to their paper. The discussion brought up by other reviewers, and addressed by the authors were also interesting.

The paper is overall of good quality, and has sufficient material for complete reproducibility and understanding.

**Limitations:**

Yes.

**Paper Formatting Concerns:**

No issue to report.

**Quality:**

4

**Strengths And Weaknesses:**

**Strengths**

+ *The paper is overall well-written*. It is easy to follow, and I found the mathematical notations quite clear.
+ *The benchmark is commendable*. The number of compared models (14) and the number of datasets (67) subdivided into 4 different categories clearly substantiate the claim of RADDT's performance. In addition, the appendix clearly contributes to enriching the results and completes well the main section.
+ *Appropriate appendices*: It turns out that most of my questions were already addressed by the appendix when reading the paper. From this criterion, I think that the appendices complement well and support the entire paper appropriately.
+ *The model is intuitive to implement*. At the end of the paper, it seems that the model can be interpreted as a typical MLP with RELU activation where each layer is twice as big as the preceding one (thus expanding the tree in breadth). This MLP has one fewer layer than the desired depth, and the softmin operator then serves as a key indicator of which prediction parameters should be retrieved from a look-up table. Though subjective, this second reading could perhaps contribute to the understanding of the performance for the authors.

**Weakneses**

Despite the mertis previously listed, few items remain unsolved in my opinion:

+ *The discussion on the number of parameters is somewhat limited*. From the introduction, the authors argue that oblique trees are "[a]iming for higher accuracy with fewer parameters", and similarly that they "tend to generate smaller trees [...] compared to orthogonal trees like CART." (Appendix K.10). While I acknowledge that an oblique decision compensates for multiple orthogonal decisions, I find that quite confusing as the depth seems to be a user-defined parameter in oblique tree, as especially shown in experiments with arbitrary depth set to 12. In addition, since all nodes use linear decision, the number of parameter seems to be $(2^D-1+2D)(d+1)$ for $d$ features and depth $D$ in a regression task, which can be quite expensive for a large number of features, or even some deep trees. Could the authors mitigate the cost of their accuracy improvements depending on the number of parameters compared to other models? In line with this comment, I believe that a discussion on the training time specifically depending on the number of samples, features and tree depth would enhance the trade-off rationale for selecting RADDT as a model in future tasks.
+ *Discussion on initialisation could be improved*. The authors acknowledge that initialisation is crucial to their decision tree, and devise local schemes based on maximum margins to between samples. However, this section (4.2) seemed insufficiently clear. The binary classes are not specified for training each SVM per node of the tree. Moreover, it seems that the corner case where all the training dataset ends up in a single unique path is not considered.
+ *Few missing references*. In this paper, authors consider only differentiability through the prism of oblique trees. Yet the literature as also shown alternative ways of definining it, e.g. [1]. In addition, I think that [2] is absent from the discussion, even though it defines *sparse* oblique trees and seem quite appropriate to the scope of the paper.
+ *Some additional discussion on interpretability would be welcome*. in line with the sugestion of [2], it seems a bit unconventional to read a paper on decision trees, yet have most of the interpretability put away in the appendix (K.10). Although the authors clearly ague that there are interpretatbility benefits in terms of "tree-based prediction, logic, unique prediction path and the complexity of decision rules" (section 5.5), it seems to obfuscate the fact that oblique rules are harder to properly interprete than orthogonal rules or sparse oblique rules. As there is a consensus that depth in trees can be harmful, I would like to understand how the authors argue that for a relatively small depth (e.g. 5) and few features (e.g. 10) the resulting model is interpretable.

**Minor comments**

+ Just two typos
	+ (l 155): zero-viloation => zero-violation
	+ (l 156): positve => positive
+ The term "generate" in Alg 1 (l12) is slightly unclear. Are the parameters $\pmb{A}_{\text{iter}}$ , etc, equal to $\pmb{A}_{N_\text{epoch}}$ , etc ?
+ An anonymised repo link would have enhanced the review by ensuring that code was available.

If the authors convince me in addressing the few points above, especially regarding interpretability, I might raise my score.

[1] Yang, Y., Morillo, I. G., & Hospedales, T. M. (2018). Deep neural decision trees. arXiv preprint arXiv:1806.06988.

[2] Carreira-Perpinan, M. A., & Tavallali, P. (2018). Alternating Optimization of Decision Trees, with Application to Learning Sparse Oblique Trees. Advances in Neural Information Processing Systems, 31.

---

> ### Author Rebuttal · Authors · 2025-07-31
>
> We sincerely thank the reviewer for the constructive feedbacks, as well as for recognizing the contributions of our work. Below, we provide point-by-point responses to address each concern raised.
>
> >**Q1-"The discussion on the number of parameters in limited... the depth seems to be a user-defined parameter in oblique trees... Could the authors mitigate the cost of their accuracy improvements depending on the number of parameters?"**
>
> Thank you for the suggestion. We will include this parameter-based comparison in the revision to better clarify model complexity.
>
> To clarify, tree depth is a key hyperparameter in all decision tree models, not specifically defined by our method. It determines the number of nodes, and thus governs the total number of parameters. While oblique trees require more parameters due to feature combinations, they can achieve similar accuracy with significantly smaller depth, meaning they do not necessarily require more parameters overall compared to axis-aligned tree CART.
>
> For a D-depth tree with $T_B=2^D-1$ branch nodes and $T_L=2^D$ leaf nodes and a dataset with p features:
> - CART requires $T_B *2 + T_L$ parameters.
> - Our GET uses $T_B * p + 1 + T_L$ parameters.
> - Our GET-Linear uses $T_B * p + 1 + T_L * p + 1$ parameters.
>
> Based on results in Tables 1 and 3, CART achieves 74.85% accuracy with average depth 10.24 and ~3,141 parameters. GET and GET-Linear reach 82.39% and 84.63% accuracy at depths 7.29 and 4.82, using ~10,504 and ~4,045 parameters, respectively. Notably, if the goal is to match or slightly exceed CART's accuracy, our models can do so with much smaller trees. Our new comparison results show that GET achieves 75.23% at depth 2.82 using only 101.71 parameters, and GET-Linear reaches 77.21% at depth 1 with just 40.47 parameters, **yielding 31× and 78× reductions in parameter count compared to CART**.
>
> Thus, **our methods do not necessarily require more parameters for improved accuracy**; the total parameter cost depends on the desired trade-off between accuracy and model complexity.
>
> |Method (Accuracy)|Depth|Number of Branch Nodes|Number of Leaf Nodes| Total Number of Parameters|
> |-|-|-|-|-|
> |CART (74.85)|10.24|1,046.65|1,047.65|3,140.94|
> |GET (82.39)|7.29|671.94|672.94|10,503.94|
> |GET-Linear (84.63)|4.82|129|130|4,044.71|
> |GET (75.23)|2.82|6.88|7.88|101.71|
> |GET-Linear (77.21)|1|1|2|40.47|
>
> >**Q2-"a discussion on the training time specially depending on the number of samples, features, and tree depth would enhance the trade-off rationale."**
>
> Thank you for your insightful comment. In Section 5.3 and Appendix K.8 and K.9, we provide a preliminary discussion, but we acknowledge this may not have been sufficiently clear. We clarify this here and will include it in the revision.
>
> Specifically, our training time is primarily affected by three factors: the number of training samples $n$, features $p$, and tree depth $D$. For D-depth tree with $T_B=2^D-1$ branch nodes, each associated with an oblique split defined by $\mathbf{a}_t \in \mathbb{R}^p$ and $b_t \in \mathbb{R}$. This results in a weight matrix $\mathbf{A} \in \mathbb{R}^{(2^D-1) \times p}$ and a threshold $\mathbf{b} \in \mathbb{R}^{(2^D-1)}$. During training, each sample must be evaluated at all branch nodes via the matrix operation $\mathbf{A}\mathbf{x}_i$，requiring $O(p \cdot (2^D-1))$ operations per sample and $O(n \cdot p \cdot (2^D-1))$ operations for the full dataset.
>
> This trend is observed in our experiments. Table 4 shows training time increases with tree depth, while Table 5 demonstrates a similar increase with sample size in our million-scale scalability study.
>
> Importantly, our training pipeline is highly parallelizable and benefits significantly from GPU acceleration.
> The advantages become more pronounced when dealing with larger matrix operations, especially for larger $n$, $p$ and $D$. Under ideal full GPU utilization, training can also be distributed across multiple GPUs, further reducing time in proportion to the number of GPUs used.
>
> >**Q3-"Discussion on initialization could be improved... The binary classes are not specified for training SVM."**
>
> Thank you for raising this important point. While we provided an example in Appendix E, Figure 3, we agree the explanation needs clarification and will revise it accordingly.
>
> Improper initialization, such as when $\mathbf{a}_j^T \mathbf{x}_i - b_j = 0 \approx 0$, places samples close to the decision boundary. This can weaken softmin approximation, as near-zero and exact-zero violations produce similar outputs, reducing contrast in routing decisions (clear 0 for non-zero violation and 1 for exact-zero). To mitigate this, $\mathbf{a}_j$ and $b_j$ should be initialized with sufficient margin to the boundary.
>
> Each branch test $\mathbf{a}_t^T\mathbf{x}_i \leq b_t$ acts as a binary classifier, directing samples left or right. A practical adjustment is to slightly shift the decision boundary (e.g., by modifying the median of $\mathbf{a}_j^T \mathbf{x}_i$) to enlarge the margin without changing sample assignments. Beyond only adjusting $b_j$, we also refine both $\mathbf{a}_j$ and $b_j$ using an SVM: **we treat left/right assignments as binary labels and fit a linear SVM to maximize margin**. The resulting weights and bias update the split parameters while preserving the original assignment. This procedure is visualized in Appendix E.
>
>
> >**Q4-"Few missing references  e.g. [1] (Yang et al., 2018) [2] (Carreira-Perpinan and Tavallali, 2018) is absent from the discussion."**
>
> Thank you for suggesting these two excellent studies. Regarding [2] of Tree Alternating Optimization (TAO) method, we have already included their regression tree with linear predictions (Zharmagambetov and Carreira-Perpinan, 2020), in our comparisons.
>
> As for [1], the Deep Neural Decision Tree (DNDT) is indeed a pioneering work in gradient-based trees. DNDT reconstructs the tree as a neural network and relies on Kronecker products for class predictions. In contrast, our method offers an exact reformulation without altering its classic tree structure. While both are gradient-based, they differ in structure, formulation and optimization. We see them as complementary work suited to different applications and will clarify this distinction and include the DNDT reference in the manuscript.
>
> >**Q5-"additional discussion on interpretability would be welcome... it seems to obfuscate the fact that oblique rules are harder to interprete than orthogonal rules. As there is a consensus that depth in trees can be harmful, I would like to understand how the authors argue that for a relatively small depth (e.g. 5) and few features (e.g. 10) the resulting model is interpretable."**
>
> Thank you for your valuable feedback on interpretability. To address your concerns, we clarify our interpretability through both illustrative examples and experimental evidence.
>
> We fully agree that **orthogonal trees like CART are more interpretable with a same depth**. However, as originally discussed in OC1 (Murthy et al., 1994), **oblique trees can be more interpretable when their required depth is significantly reduced**. For example, as shown in Figure 7, on a synthetic 2-feature dataset, our method achieves comparable accuracy with a depth-4 tree (16 rules), whereas CART requires depth-9 (512 rules). In such cases, the simpler structure outweighs the complexity of oblique splits.
>
> Moreover, as in our response to your Q1, our GET and GET-Linear achieve slightly better accuracy than CART while using smaller trees, average depths of 2.82 and 1.0 compared to CART's 10.24, **resulting in 30–70× fewer parameters**. For example, *a depth-1 tree consists of just two rules and remains highly interpretable*: "IF the linear combination of features is below a threshold, THEN predict from the left leaf; OTHERWISE, predict from the right."
>
> Regarding your example of a depth-5 tree with 10 features, we agree that an orthogonal tree is easier to interpret at the same depth. However, our method typically achieves similar or better accuracy with far smaller depth, **reducing model parameters and rule count**.
>
> >**Q6-"Minor Comments"**
>
> >Q6-1 "the term "generate" in Alg 1 is unclear. Are $\mathbf{A}_{\text{iter}}$ equal to $\mathbf{A}_{N_\text{epoch}}$?."
>
> Thank you for careful review of our algorithm. We apologize for the confusion and would like to clarify it.
>
> The term "generate" indicates that a tree candidate can be constructed based on the parameters obtained after $N_\text{epoch}$ training. Specifically, $\mathbf{A}_{\text{iter}} = \mathbf{A}_{N_\text{epoch}}$ and $\mathbf{b}_{\text{iter}} = \mathbf{b}_{N_\text{epoch}}$, while $\theta_{\text{iter}}$ is derived through a deterministic recalculation and may not strictly equal $\theta_{N_\text{epoch}}$.
>
> As described in Section 3.2, once $\mathbf{A}_{N_\text{epoch}}$ and $\mathbf{b}_{N_\text{epoch}}$ are obtained, the deterministic path of each sample can be computed, allowing us to identify sample-to-leaf assignments. For regression, $\theta_t$ is recalculated as the average of the true targets $y_i$ assigned to that leaf.
> We will revise the manuscript to clarify this explanation.
>
> >Q6-2 "An anonymised repo link would have enhanced the review by ensuring that code was available."
>
> Thank you for your interest in the code. We have indeed provided the complete code in a GitHub repository. The link is given in the last line of Page $2$ of the manuscript. We will also ensure it is publicly accessible upon acceptance.

---

> > ### Comment · Reviewer_oByt · 2025-08-02
> >
> > I would like to thank the authors for answering my few comments, uncertainties, and pointing out the elements that I might have missed. I appreciate the additional results and insights brought here. In light of all elements, I chose to maintain my score. I believe that the manuscript has become clearer given those answers.
> >
> > I do have a few other minor comments that are of no relevance towards the final rating, but only there for the sake of discussion.
> > - I acknowledge the authors' perspective on interpretability, as well as the provided reference. However, despite the rationale, it seems to me that, in absence of sparsity, a chain of even few linear regressions for node splitting in oblique tree is hard to interpret. Of course, the authors agreed that axis-aligned trees are easier to interpret in general under reduced depth. As this turns to be closer to an issue of trading accuracy for interpretability, let alone the discussion in number of parameters, I consider that the argument was addressed because that discussion could go beyond RADDT.
> > - I slightly disagree with the authors regarding the depth hyperparameter. For some tree models, e.g. CART; the depth is considered to be a maximum, and in this sense is a purely optional hyperparameter. That is even the default in scikit-learn, as the authors mentioned to reviewer #ZgpM. In RADDT, it is mandatory, meaning that its impact requires further control or understanding.

---

> > > ### Author Response · Authors · 2025-08-04
> > >
> > > We sincerely appreciate your time, thoughtful evaluation, and recognition of our work, as well as the valuable discussion provided throughout the review process. Thank you especially for your additional insights on interpretability and the role of tree depth, which have further deepened our understanding of these important aspects.
> > >
> > > We fully agree with your perspective that the interpretability of oblique trees involves multiple considerations beyond depth and rule count, including sparsity, split type, and the trade-off between accuracy and simplicity, as you noted. Interpretability is not tied to a specific tree type but is shaped by a combination of factors. We will revise the manuscript to clarify this point and present a more nuanced discussion.
> > >
> > > We also appreciate your comment on the role of tree depth in our method, particularly in comparison to classic models like CART, where depth is treated as optional with a default maximum value. We will incorporate this clarification in the revision to highlight the distinction and enhance understanding.
> > >
> > > Thank you once again for your constructive suggestions and thoughtful engagement. Your insights have contributed meaningfully to improving the clarity and depth of our manuscript.

---

### Official Review · Reviewer_ZgpM · 2025-06-20

**Clarity:** 3
**Significance:** 3
**Originality:** 3
**Rating:** 5
**Confidence:** 5

**Summary:**

This paper introduces a novel differentiable decision tree method called RADDT, leveraging a "ReLU+Argmin" formulation. The key contributions include addressing non-differentiability issues through ReLU functions combined with an approximated argmin operation via scaled softmin. Additionally, the authors introduce a multi-run warm-start annealing strategy to enhance training stability. Empirical results demonstrate substantial accuracy improvements compared to baseline methods.

**Questions:**

Q1: [3] proposes a method to handle soft decisions in decision trees. What advantages does your method have over [3], and why was this comparison omitted?

Q2: What advantages does your method provide compared to selecting the path with maximum probability, as proposed by [4]?

Q3: RADDT significantly outperforms standard SDTs. What happens when the ReLU+Argmin approach is removed, and a standard STD is trained using your advanced training protocol?

Q4: Could an entmax-like transformation be utilized for your softmin operation to ensure smoother gradients?

---

[3] Norouzi, Mohammad, et al. "Efficient non-greedy optimization of decision trees." Advances in Neural Information Processing Systems 28 (2015).

[4] Frosst, Nicholas, and Geoffrey Hinton. "Distilling a neural network into a soft decision tree." arXiv preprint arXiv:1711.09784 (2017).

**Ethical Concerns:**

["NO or VERY MINOR ethics concerns only"]

**Final Justification:**

This paper presents a novel differentiable decision tree method based on a ReLU+argmin formulation, combined with an annealing training strategy, and shows promising empirical results along with open-sourced code. The authors provided a detailed and thoughtful rebuttal that addressed the most critical issues raised in the initial review. In particular:

- The confusion between axis-aligned and oblique trees was clarified, and the authors agreed to make this distinction explicit in the revised paper.
- Concerns regarding hyperparameter optimization (HPO) were addressed with extensive tuning experiments. While I believe more recent tuning practices (e.g., for XGBoost) could still be adopted, the current results are sufficient to support the paper's claims.
- The interpretability discussion was expanded and contextualized better, particularly with respect to depth and rule complexity.
- The authors acknowledged additional prior work and agreed to include appropriate citations and clarifications.

Overall, the revised paper will be significantly strengthened, and I believe it can make a contribution to the field.

**Limitations:**

No, see weaknesses (specifically regarding interpretability).

**Paper Formatting Concerns:**

No concerns.

**Quality:**

4

**Strengths And Weaknesses:**

## Strengths

- **Novel ReLU Formulation**: The use of ReLU and argmin for differentiable decision trees is innovative and, to the best of my knowledge, novel.
- **Multi-run Annealing Strategy**: The proposed annealing method appears novel and shows a significant impact on performance in the ablation study.

---

## Weaknesses

### Major Weaknesses

- **Confusion Between Axis-Aligned and Oblique Trees**:
  - The paper mixes standard (axis-aligned) and oblique decision trees, potentially confusing readers.
  - Comparisons with CART, RF, and GradTree (especially in the abstract) lack clear explanation of fundamental differences. Performance gains over CART are unsurprising given the significantly higher complexity of oblique splits.
  - Tables only categorize by optimization method, obscuring the types of decision trees compared (axis-aligned vs. oblique). This is particularly problematic when comparing complexity in terms of tree depth (Tables 1 and especially 2). Axis-aligned DTs consider only one feature per split, whereas oblique DTs consider linear combinations of features. For instance, the ailerons dataset has 40 features, significantly affecting complexity and making tree depth a poor proxy. An alternative measure, such as parameter count, should be added. Additionally, CART trees are typically not fully-grown to their maximum depth, a factor not currently considered.
  - The abstract, introduction, and conclusion emphasize interpretability of standard trees, yet the proposed method is oblique without adequately discussing interpretability implications.
  - The scenarios presented in Appendix A not only argue in favor of hard decisions, but also—at least implicitly—support axis-aligned splits. For example, the discussion of pressure exceeding a threshold clearly suggests a univariate (i.e., axis-aligned) split, which cannot be directly modeled using oblique splits. This stands in contrast to the proposal of hard yet oblique decision trees, and again introduces confusion.

- **Improper Hyperparameter Optimization (HPO)**:
  - Hyperparameters for baseline models (CART, RF, especially XGBoost) are not adequately tuned, violating standard scientific practice for tabular data.
  - Claims of beating XGBoost without thorough HPO are misleading.

- **Interpretability Analysis**:
  - I strongly disagree with the interpretability analysis in its current form. IF-THEN rules typically mirror human decision-making closely (see e.g. [1]). Oblique trees have inherently lower interpretability at individual splits, especially when not sparse—and no sparsity is demonstrated. Making interpretability claims without proper analysis, particularly on real-world datasets, is highly problematic.

### Minor Weaknesses
  - In Section 4.3, the phrase "optimal solutions" could be misunderstood in the context of optimal decision tree literature; a more careful wording is suggested.
  - Alternative metrics such as ROC AUC, MAE, and MSE are missing. Accuracy alone is insufficient; including these (at least in the appendix) would strengthen the evaluation.
  - The authors use reinforcement learning as an example where non-differentiability remains a major barrier, but methods already exist addressing this issue using gradient-based decision trees (see [2]).

---

While my review may come across as quite critical, this is directed at the current state of the paper rather than the underlying idea, which I find promising. If the authors are able to address the concerns raised, I would be happy to revise my score. However, in its present form, I see several substantial issues.

---

[1] Hastie, Trevor, Robert Tibshirani, and Jerome Friedman. "The elements of statistical learning." (2009).

[2] Marton, Sascha, et al. "Mitigating information loss in tree-based reinforcement learning via direct optimization." (2025).

---

> ### Author Rebuttal · Authors · 2025-07-31
>
> We sincerely thank the reviewer for their effort in evaluating our work and for the thought-provoking questions. We address each point in detail below.
>
>
> > **Q1-"Confusion between axis-aligned and oblique trees"**
>
> > Q1-1 "Comparison with CART RF and GradTree lacks explanation. Performance gains over CART are unsurprising..."
>
> Thank you for pointing this out. We appreciate the suggestion and will include the clarification in the revision.
>
> Our primary focus is on comparing oblique trees. Among the 14 baselines, all are oblique except for CART, RF, XGBoost, and GradTree. We include CART as a widely recognized foundational method. Its inclusion aligns with prior work on oblique trees, such as TAO (Zharmagambetov and Carreira-Perpinan, 2020), DGT (Karthikeyan et al., 2022), DTSemNet (Panda et al., 2024), TEL (Hazimeh et al., 2020), which also compare against CART. Although outperforming CART is expected, its inclusion helps quantify the amount of improvements achievable by oblique tree.
>
> GradTree is included as a representative gradient-based tree using STE, enabling fair comparison against other STE-based oblique trees like DGT and DTSemNet.
>
> Since it remains uncertain whether oblique trees can match ensemble's accuracy, we include RF as a commonly used and strong ensemble baseline, and TEL as the oblique ensemble counterpart in our comparisons.
>
> > Q1-2 "Alternative parameters count, should be added."
>
> Thank you for valuable suggestion. We agree that parameter count offers a clearer view of model complexity. While oblique tree typically require more parameters at a given depth, it often yields better accuracy with much smaller depth. **Thus, for similar accuracy, oblique tree may not require more parameters overall**.
>
> For a D-depth tree with $T_B=2^D-1$ branch nodes and $T_L$ leaf nodes and p input features:
> - Axis-aligned tree requires $T_B *2 + T_L$ parameters.
> - Our GET uses $T_B * p + 1 + T_L$ parameters.
> - Our GET-Linear uses $T_B * p + 1 + T_L * p + 1$ parameters.
>
>
> From Tables 1 and 3, CART achieves 74.85% accuracy with average depth 10.24 and ~3,141 parameters. GET and GET-Linear reach 82.39% and 84.63% accuracy at depths 7.29 and 4.82, using ~10,504 and ~4,045 parameters, respectively. Notably, if the goal is to match or slightly exceed CART's accuracy, our models can do so with much smaller trees. Our new experiments show that GET achieves 75.23% at depth 2.82 using only 101.71 parameters, and GET-Linear reaches 77.21% at depth 1 with just 40.47 parameters, **yielding 31× and 78× reductions in parameter count compared to CART**.
>
> |Method (Accuracy)|Depth|Number of Branch Nodes|Number of Leaf Nodes|Total Number of Parameters|
> |-|-|-|-|-|
> |CART (74.85)|10.24|1,046.65|1,047.65|3,140.94|
> |GET (82.39)|7.29|671.94|672.94|10,503.94|
> |GET-Linear (84.63)|4.82|129|130|4,044.71|
> |GET (75.23)|2.82|6.88|7.88|101.71|
> |GET-Linear (77.21)|1|1|2|40.47|
>
> > Q1-3 "CART are not fully-grown to maximum depth."
>
> Thank you for highlighting this point. A fully-grown CART, expanded until all leaves are pure as in default sklearn setting, are rarely used in practice due to overfitting and reduced accuracy. In this regard, we tune CART's depth from 1 to 100. This 100-depth search is sufficient to identify its optimal depth. For reference, a fully-grown CART leads to an average test accuracy drop of 9.27% compared to our tuned version reported in Table 1.
>
> >**Q2-"Improper HPO. Hyperparameters are not adequately tuned"**
>
> Thank you for your thoughtful comments. We would like to clarify that we have carefully tuned hyperparameters for RF and XGBoost, as detailed in Section 5.1 and Appendix J. Our tuning follows the guidelines from Oshiro et al. (2012) and Probst et al. (2019), which emphasize the importance of tuning the number of trees and tree depth. We tune the number of trees from {50, 100, 200, 300, 400, 500} and depth from 1 to 50, while keeping other parameters at default values as recommended.
>
> This setup yields 300 tuning combinations for both RF and XGBoost. In contrast, our method is tuned only over tree depth from 1 to 12 (12 combinations), ensuring a much lighter tuning for fairness. To further address your concern, we extend the search to 5,400 combinations: depth in {1–50, 55, 60, 70, 80, 90, 100, 150, 200, 250, 300}，number of trees in {5, 10, 50, 100, 200, 300, 400, 500, 600, 700}， with additional tuning for RF (min samples per leaf and per split in {1, 2, 4}), and for XGBoost (subsample and colsample_bytre in {0.6, 0.8, 1.0}).
>
> These 5,400-combination yield slight improvements: +0.11% for RF and +0.26% for XGBoost over the 300-combination. We further use PyCaret library to perform an extensive 10,000-combination HPO for XGBoost across all its hyperparameters, leading to just a 1% gain. A summary is given below:
>
> |Method|Tuning Combinations|Accuracy|
> |-|-|-|
> |RF|300|82.62|
> |RF|5,400|82.73|
> |XGBoost|300|83.51|
> |XGBoost|5,400|83.77|
> |XGBoost|10,000 (by PyCaret)|84.55|
> |Our GET|12|82.39|
> |Our GET-Linear|12|84.63|
>
> Despite extensive tuning, our key finding remains unchanged: our single tree achieves accuracy comparable to ensemble methods. The statistical Friedman rank further shows no significant difference, as XGBoost shares the top rank with ours.
>
> Importantly, our aim is not to claim superiority over ensembles. Since any base learner can be improved via ensembling like bagging, comparing a single tree to ensembles is inherently unfair. Instead, we aim to provide a reference showing that our tree can match ensemble's accuracy with far fewer parameters.
>
> >**Q3-"Interpretability"**
>
> Thank you for your insightful comments. We fully agree that, at the same depth, axis-aligned trees like CART are more interpretable. However, as originally noted in OC1 (Murthy et al., 1994), oblique trees could become more interpretable when their depth is substantially smaller.
>
> For example, as shown in Appendix K10, on a synthetic 2-feature dataset, our method yields a depth-4 oblique tree with 16 rules, whereas CART requires depth-9 with 512 rules. Despite using 2-feature combinations, the much smaller depth improves interpretability. Similarly, as noted in Q1-2, our models can also achieve slightly better accuracy than CART with  much smaller depths: 2.82 (GET) and 1 (GET-Linear) compared to 10.24 for CART, leading to 30–70× fewer parameters.
>
> >**Q4-"minor weaknesses"**
>
> > Q4-1 "Alternative metrics such ROC AUC are missing."
>
> Thank you for your suggestion on additional metrics. For same-dataset comparison on Group (ii) datasets, we adopt the same metrics used in the original works we compared with: RMSE for DTSemNet, DGT-Linear, and TAO-Linear, and F1-score for GradTree. For other experiments, we mainly report accuracy and Rank. We agree that additional metrics can offer a more comprehensive evaluation and will incorporate them in the revision.
>
> > Q4-3 "reinforcement learning... but method already exists like (Marton et al., 2025)."
>
> Thank you for bringing this excellent study to our attention. We will cite it in the revision. It adopts GradTree (Marton et al., 2023) to learn decision tree policies, a method we already included in our comparisons.
>
> While both methods are gradient-based, they differ in formulation and optimization. Their approach uses axis-aligned trees with STE, offering efficiency for certain tasks, whereas ours focuses on oblique trees with differentiable approximations, yielding strong empirical performance. We view these two work are complementary with different applicability, and will clarify this distinction in the revision.
>
> >**Q5-"Other questions"**
>
> > Q5-1 "(Norouzi et al., 2015) proposes a method to handle soft decisions. What advantages does your method have, and why was this comparison omitted?"
>
> Thank you for bringing this important work to our attention. Norouzi et al. (2015) introduce a surrogate loss to enable optimization. In contrast, our method uses an exact reformulation of classic tree with a different optimization. These represent two distinct approaches. Due to the lack of public code and the table-based results in their paper, a direct comparison was not feasible. We will cite this work and clarify the distinction in the revision.
>
> > Q5-2 "What advantages do your method provides over SDT (Frosst and Hinton, 2017)? What happens if a SDT is trained using your advanced training protocol?"
>
> Thank you for your insightful comments. We have included SDT in our manuscript. SDT uses a sigmoid function to approximate hard decisions, resulting in probabilistic paths. However, these approximations accumulate across all nodes, potentially degrading accuracy in deeper trees. In contrast, our method is based on an exact reformulation. Its superiority can be attributed to our loss formulation and optimization strategies like annealing.
>
> Regarding applying our training protocol to SDT, we agree this is an interesting direction. However, effects are difficult to predict without extensive experiments. While our strategies may help mitigate approximation issues for smaller trees, deeper trees may still suffer from accumulated errors. For instance, even with a strong sigmoid output of 0.9 per node, the path probability at depth 10 becomes $0.9^{10}=0.35$ (in worse case of $0.55^{10}=0.003$), leading to poor sample assignments. In our current experiments, we use the original SDT code from GitHub without modification for fair comparison. We will further explore applying our training strategies to SDT and include this in the revision.
>
> > Q5-3 "Could an entmax be utilized for your softmin operation?"
>
> Thank you for this thoughtful suggestion. Entmax, as used in GradTree (Marton et al., 2023), generalizes softmax by allowing sparse outputs controlled by a scalar, recovering softmax at 1 and producing exact zero at 2. Given its similarity to our scaled softmin, we agree that entmax is a promising alternative. However, selecting a sparsity parameter is challenging. We will explore this and discuss it in the revision.

---

> ### Comment · Reviewer_ZgpM · 2025-08-01
>
> I would like to thank the authors for their detailed and thoughtful rebuttal. I appreciate the effort taken to address the raised concerns. Most importantly, I believe the authors have adequately clarified the confusion between axis-aligned and oblique trees and agreed to make this distinction clearer in the revision. Given this, I will adjust my rating accordingly and now suggest acceptance (4 - borderline accept). I believe the revised paper will be significantly improved and constitutes a valuable contribution, particularly with the open-sourcing of the code.
>
> I do have a few additional minor comments that did not influence my updated rating, as I do not see them as blocking acceptance:
>
>   - **Regarding fully-grown CART trees:** My original comment was not about using fully-grown trees in practice, but rather about the fact that *maximum depth* can be misleading as a proxy for complexity, especially since CART is typically pruned. I fully agree with the authors' clarification here and appreciate their depth tuning approach.
>
>   - **On hyperparameter optimization (HPO):** While the effort to tune XGBoost and RF is appreciated, I would like to note that recent literature suggests that tuning the number of trees (estimators) is not necessary if early stopping is used. Instead, regularization parameters (including `reg_alpha` and `reg_lambda`) play a more crucial role in the performance of GBDTs, specifically XGBoost. I recommend using predefined tuning grids based on recent work such as [1,2,3]. While I still believe the tuning presented in the paper is sufficient to support the claims (especially since the goal is not to claim SOTA performance), I think this could be improved to strengthen the empirical evaluation.
>
>   - **Comparisons with more recent models:** It would be interesting to see comparisons with additional recent models for tabular data such as CatBoost, TabM, or RealMLP. However, I consider these suggestions for future work and not necessary for acceptance.
>
> [1] Borisov, Vadim, et al. “Deep neural networks and tabular data: A survey.” IEEE TNNLS, 2022.
> [2] Grinsztajn, Léo, et al. “Why do tree-based models still outperform deep learning on typical tabular data?” NeurIPS, 2022.
> [3] McElfresh, Duncan, et al. “When do neural nets outperform boosted trees on tabular data?” NeurIPS, 2023.

---

> > ### Author Response · Authors · 2025-08-02
> >
> > We sincerely thank the reviewer for the thoughtful reassessment and the encouraging score increase. We greatly appreciate your recognition of our work and the constructive feedback provided throughout the review process. Your insightful comments on axis-aligned trees and HPO have helped us significantly improve the clarity of our manuscript.
> >
> > In the revision, we will better highlight the distinction between axis-aligned and oblique trees and incorporate parameter count as an alternative proxy for model complexity, as you suggested. We will also include additional experiments using the recommended tuning grids, such as regularization parameters, and add comparisons against recent tabular-data models.
> >
> > We are also grateful for your suggestions on promising directions and valuable baselines for future research. These ideas are highly relevant and will be explored in our future work.
> >
> > Once again, we deeply appreciate your time and responsible reviewing. Your feedback has played a crucial role in enhancing the quality of our paper, and we look forward to submitting a revised version that reflects these improvements and contributes meaningfully to the field.

---

> > > ### Comment · Reviewer_ZgpM · 2025-08-04
> > >
> > > Thank you for the additional clarification. After revisiting the rebuttal and the paper, I am inclined to increase my score to a 5 to more clearly reflect my belief that the paper should be accepted. I look forward to seeing the revised version incorporating the suggested improvements.

---

> > > > ### Author Response · Authors · 2025-08-04
> > > >
> > > > Thank you for your recognition of our work and for raising the score. We are pleased to hear that our clarifications were helpful, and we will certainly incorporate the suggested improvements into the revised version. Your thoughtful feedback has been instrumental in improving the clarity and quality of our paper, and we sincerely appreciate your support throughout the review process.

---

### Official Review · Reviewer_pLD8 · 2025-07-02

**Clarity:** 3
**Significance:** 3
**Originality:** 3
**Rating:** 5
**Confidence:** 3

**Summary:**

- This paper proposes a gradient-based learning method for oblique decision trees.
- It first exactly reformulates the problem of learning hard-split oblique trees into an unconstrained optimization problem, minimizing a loss function constructed using the ReLU function and the argmin operator. Next, it introduces a surrogate loss by replacing the argmin operator with a differentiable softmax function, and then learns the tree using a gradient-based method.
- Numerical experiments demonstrate that the proposed method yields decision trees with higher test accuracy compared to existing gradient-based decision tree learning approaches.

**Questions:**

- Regarding the first weakness I raised: Does the proposed method still achieve better accuracy on the training data compared to existing methods, even without using the warm-start annealing strategy for both? Please point out if I've overlooked any experimental results addressing this.
- Concerning the ReLU-based cumulative violation: If $a_j$ and $b_j$ are  zero for all $j$, then $v_{i,j}$ is zero for all $j$. In this specific case, the unique path might not be determined by Eq. (4). While this may be a corner case, I am interested in whether such a scenario could cause any practical or numerical issues during the training process.
- The motivation for using simulated annealing to adjust $\alpha$ isn't entirely clear to me. As $\alpha$ increases, the surrogate loss (6) approaches (5), so from the perspective of minimizing (5), it would be reasonable to set $\alpha$ as large as possible (assuming numerical stability issues don't arise). From this viewpoint, couldn't a binary search-like approach be more efficient for adjustment than simulated annealing?

**Ethical Concerns:**

["NO or VERY MINOR ethics concerns only"]

**Final Justification:**

The authors have sincerely addressed all of my concerns and questions. While I recommend they revise the manuscript to incorporate these points, I believe the paper is suitable for publication.

**Limitations:**

yes

**Quality:**

3

**Strengths And Weaknesses:**

- Strengths
	- Decision trees are widely used as highly interpretable predictive models, and the proposed computationally efficient learning algorithm for decision trees holds significant value in the field of machine learning.
	- The research motivation is clear, and the proposed method is explained in an understandable manner.
	- The computational experiments are comprehensive. Notably, the validation of effectiveness through variance testing is commendable. Furthermore, it's interesting that the optimization in training leads to improved accuracy compared to existing gradient-based learning methods.

- Weaknesses
	- The proposed method's performance is significantly improved by a multi-run warm-start annealing strategy. While this isn't inherently problematic, it makes it difficult to ascertain whether the effectiveness of the proposed method, relative to prior gradient-based decision tree learning research, is fundamentally due to the ReLU-based reformulation itself.
	- The numerical experiments show that the proposed method can obtain trees with lower loss on training data than existing gradient-based approaches. However, the theoretical insights from a mathematical optimization perspective as to why such results are obtained are somewhat limited.

---

> ### Author Rebuttal · Authors · 2025-07-31
>
> We sincerely appreciate the reviewer's recognition of our work, as well as the constructive feedback provided. These comments are highly valuable and have helped us improve the clarity and completeness of our manuscript. Below, we carefully address each of the concerns and questions raised.
>
>
> >**Q1-"Regarding the first weakness I raised: Does the proposed method still achieve better accuracy... even without the warm-start annealing strategy? Please point out if I've overlooked any experimental results addressing this."**
>
> Thank you for your important attention to the annealing strategy. In our manuscript, we provide an ablation study demonstrating its effectiveness at the end of Section 5.2 and in Appendix K.7.
>
> Without the annealing strategy, training accuracy drops significantly, especially when using the standard softmin without any scaling (i.e., $\alpha=1$). Specifically, as reported in Appendix K.7, Table 19. Training accuracy with $\alpha=1$ is 9.7%, 9.32%, and 8.46% lower than with annealing at tree depths of 2, 4, and 8, respectively. This decline is due to the large approximation gap introduced by the standard softmin function, as illustrated in Appendix F, Figure 6.
>
> Even without annealing, the issue can be partially mitigated by increasing the softmin scale factor $\alpha$. This intuitive adjustment does increase performance; for instance, setting $\alpha = 150$ improves training accuracy by 7.22%, 8.34%, and 6.62% over $\alpha = 1$ at the respective depths. However, it still underperforms compared to annealing, likely because $\alpha = 150$ is not optimal. Increasing $\alpha$ helps to reduce this gap, but excessively large values can lead to numerical instability. Therefore, simply increasing $\alpha$ is not a reliable solution. For reference, the results from the manuscript are summarized below:
>
> |  | D=2|D=4|D=8|
> |-|-|-|-|
> |$\alpha=1$ (Standard Softmin Function)|60.78|69.97|81.46|
> |$\alpha=150$|68.00|78.31|88.08|
> |With Our Annealing|70.48|79.29|89.92|
>
> Therefore, scaling the softmin with an appropriately chosen $\alpha$ can improve accuracy even without the annealing strategy. However, identifying the optimal $\alpha$ is challenging. Our annealing strategy is specifically designed to address this challenge by dynamically balancing approximation quality and stability throughout training. We will further discuss its effectiveness in response to your Q3 below.
>
> >**Q2-"Concerning the ReLU-based cumulative violation: If $a_j$ and $b_j$ are zero... while this may be a corner case, I am interested in whether such a scenario could cause any practical or numerical issues during the training process."**
>
> Thank you for your insightful comment on this corner case. Indeed, if both $\mathbf{a}_j = \mathbf{0}$ and $b_j = 0$, it provides same violations for all leaves, which could impair the training process.
>
> We address a similar case in Section 4.2, where we discuss the impact of initialization and introduce an adjustment strategy to handle cases where $\mathbf{a}_j^T \mathbf{x}_i - b_j = 0 \approx 0$. However, in the specific scenario you raise, when $\mathbf{a}_j = \mathbf{0}$ and $b_j = 0$, this adjustment also becomes ineffective. We acknowledge that this is a limitation of the current implementation. To mitigate this, we typically use random initialization to avoid such worst-case scenarios.
>
> Fortunately, based on our extensive experiments, such cases appear to be extremely rare in practice, and we have not observed noticeable performance degradation due to this issue. We appreciate you pointing it out and will explicitly mention this limitation in the revised manuscript.
>
> >**Q3-"The motivation for using simulated annealing to adjust $\alpha$ isn't entirely clear to me. AS $\alpha$ increases, the surrogate loss (6) approaches (5), it would be reasonable to set $alpha$ as large as possible (assuming numerical stability issues don't arise). From this viewpoint, couldn't a binary search-like approach be more efficient for adjustment than simulated annealing?"**
>
> Thank you for your valuable comment regarding the annealing strategy. Following our response to your Q1, we would like to clarify the motivation behind using annealing for adjusting $\alpha$.
>
> Binary search provides a more structured way to explore different $\alpha$ values for scaling the softmin approximation. Unlike random trials, it narrows the search range by evaluating performance at the midpoint and adjusting boundaries accordingly. While this is more principled than random guessing, it still involves testing isolated $\alpha$ values. As discussed in the response to your Q1, this approach does not match the performance of our annealing strategy.
>
> While we agree that a binary search-like approach can be used to identify an effective $\alpha$, our empirical results show that gradually increasing $\alpha$, rather than directly using a large fixed value, leads to better training outcomes. This is because starting with a smaller $\alpha$ allows the model to optimize under a smoother case, improving stability. As $\alpha$ increases, it becomes sharper and closer to the ideal formulation, and the model benefits from being progressively adapted to this more accurate approximation. Our annealing strategy takes advantage of this effect by warm-starting each successive optimization task with the solution from the previous task with smaller $\alpha$. This gradual refinement enhances the approximation accuracy, while mitigating numerical instability typically associated with larger $\alpha$.
>
> To further address your concern, we conducted a direct comparison between our annealing strategy and a binary search, using the same number of optimization runs and the same $\alpha$ search range. Specifically, our annealing strategy performs 5 optimization tasks, each with a different $\alpha$ value sampled in log-space within a predetermined range. For a fair comparison, the binary search also performs 5 sequential optimizations, where each $\alpha$ is selected via bisection within the same range. The binary search is implemented by starting at the midpoint of a given range as the initial scaling factor for the optimization. It then iteratively narrows the range by evaluating performance at the midpoint and updating one boundary accordingly. This process continues for five iterations, resulting in five optimization runs with five successive bisections.
>
> The average training accuracy across 17 regression datasets for different tree depths is summarized below. Our annealing strategy consistently outperforms binary search, with an average improvement of 2.82% across all depths.
>
> |Depth|2|4|8|12|
> |-|-|-|-|-|
> |Binary Search|69.95|77.33|87.69|94.98|
> |Our Method|71.78|82.18|90.93|96.36|
>
> These results empirically validate the effectiveness of our annealing strategy in practice. However, we acknowledge that this finding is based on empirical observation, and a theoretical analysis is still lacking. We will include this discussion and its limitation in the revised manuscript.

---

> > ### Comment · Reviewer_pLD8 · 2025-08-04
> >
> > Thank you for your detailed clarification in the rebuttal.
> >
> > Regarding your replies to Q1 and Q2, the empirical results convincingly demonstrate that adjusting $\alpha$ using simulated annealing is more effective at improving prediction performance than a binary search. Your explanation and empirical evaluation for this approach have clearly clarified its motivation and effectiveness.
> >
> > Furthermore, I appreciate the clarification provided for Q2. When you revise the manuscript, I recommend mentioning that while random initialization empirically helps to avoid the solution $\boldsymbol{a}_j, b_j = (\boldsymbol{0},0)$, there is still room for further investigation into this issue.
> >
> > Overall, since you have addressed all of my questions, I have raised my score to accept.

---

> > > ### Author Response · Authors · 2025-08-04
> > >
> > > Thank you for your time, thoughtful feedback, and recognition of our work. We are pleased that our clarifications and empirical results have effectively addressed your concerns regarding the annealing strategy and initialization. Your suggestion to highlight the potential for further investigation into the random initialization issue is greatly appreciated. We will incorporate this point into the revised manuscript to offer a more complete discussion of this limitation and future directions.
> > >
> > > Once again, thank you for your constructive comments and continued support, which have meaningfully contributed to improving the clarity and quality of our work.

---

### Public Comment · ~Kartikay_Agrawal1 · 2025-12-05
**Interesting solutions to learning scalable neural decision trees**

I came across your poster at today's event. Congratulations on acceptance of the paper. The detailed study and the time you spent on making it work for multi-class classification and regression is commendable. My team and I worked with neural decision trees trainable in a similar setup but couldn't scale it up beyond binary classification. I would be glad if you could cite it.

Thanks for your contribution. PFA.

https://ieeexplore.ieee.org/document/10798396

---

### Public Comment · ~Subrat_Prasad_Panda1 · 2026-05-07
**Incomplete characterization of DTSemNet**

Thanks for the interesting work. However, I believe the discussion and categorization of DTSemNet is incomplete and potentially misleading.

The paper groups DTSemNet with DGT and GradTree as an "STE-based" method. While this is partially true for the regression setting, it misrepresents DTSemNet's classification, which is entirely approximation-free. DTSemNet's classification uses the same core idea as the proposed RADDT: ReLU activations to encode hard branching decisions, followed by a discrete selection operation (argmax in DTSemNet vs. argmin in RADDT). The two formulations differ only in sign: DTSemNet accumulates satisfied branching activations and selects via argmax (max-pooling), while RADDT accumulates violations and selects via argmin. This is essentially the same ReLU-based encoding of hard decision trees, and neither requires STE for classification.

Although there are differences in the training procedure (e.g., annealing, softmin), given the close relationship between the two formulations, a more nuanced discussion acknowledging DTSemNet's approximation-free classification would have been appropriate.

---

### Decision · Program_Chairs · 2025-09-17

**Decision:**

Accept (spotlight)

**Comment:**

The paper proposes an exact reformulation of hard-split oblique decision trees as an unconstrained objective built from ReLU terms and an argmin operator. Next it replaces argmin with a scaled softmin surrogate to enable end-to-end gradient training. A multi-run warm-start simulated-annealing schedule on the softmin scale is introduced to bridge the surrogate and true objective gap. Experiments across many datasets and depths (up to 12) show higher test accuracy than prior gradient-trained trees, and strong scalability with GPU parallelization. A link to the code is provided in the submitted pdf. Reviewers found the approach sound and empirically strong, with the method outperforming optimal/soft trees on large datasets. The authors perform ablations which show the usefulness of their proposed annealing schedule in achieving good local optima.

The ReLU+argmin reformulation (with a differentiable softmin surrogate) is viewed as original and well-motivated and the method is explained clearly. Broad and comprehensive comparisons are performed against tree baselines with variance testing. Solid accuracy gains and speedups are obtained on sizable problems. The results scale with depth, sample size, and GPU parallelism. The warm-start annealing schedule seems empirically effective. There are some concerns about insufficient hyperparameter optimization for strong baselines (e.g., XGBoost), and reliance on accuracy without additional more fine-grained metrics. Also, the authors should tone down the interpretability claims as they learn oblique trees instead of axis-aligned trees as suggested by multiple reviewers. Other than that the authors seem to give detailed and satisfactory responses to most points raised by the reviewers.

The paper is a clean, original and empirically overall strong contribution, and there is broad consensus for acceptance. Most issues may be addressed in a revision, but it is not clear if the gains will be as impressive (e.g. with more optimized baselines or across other metrics), so I am hesitant for nominating for an oral. Also while the paper seems to be clean, relatively thorough and likely impactful (reasons for spotlight nomination), I could see technical depth of contribution as a reason for bumping down to a poster.